# Quant-dLLM: Post-Training Extreme Low-Bit Quantization for Diffusion Large Language Models

**Tianao Zhang**[1,2*]**, Zhiteng Li**[1*]**, Xianglong Yan**[1]**, Haotong Qin**[3]**, Yong Guo**[4]**, Yulun Zhang**[1†]

[1]Shanghai Jiao Tong University,  [2]Zhiyuan College, SJTU,  [3]ETH Zürich,  [4]Huawei

## Abstract

Diffusion large language models (dLLMs), which offer bidirectional context and flexible masked-denoising generation, are emerging as a compelling alternative to autoregressive (AR) LLMs. However, like AR LLMs, their model sizes continue to grow, motivating weight compression for deployment. Although post-training quantization (PTQ) is effective for AR LLMs, directly transferring it to dLLMs at 2-bit leads to unsatisfactory performance. To tackle these challenges, we propose Quant-dLLM, an ultra-low-bit PTQ framework tailored to dLLMs. Since masked-denoising activations in dLLMs differ from the fully visible signals assumed by standard PTQ methods, we introduce Masked Calibration Simulation (MCS) to align calibration with the timestep-dependent masking, which yields more reliable calibrations. Moreover, we propose a Data-aware Any-order Quantizer (DAQ) that learns ultra-low-bit weight representations via an optimization algorithm. It performs iterative approximation guided by our simulated calibration data. In addition, under a strict 2-bit budget, we introduce Adaptive Blockwise Mixed Precision (ABMP), a sensitivity-based precision allocation scheme that adaptively assigns bit width across channel groups. When restricted to 2-bit precision, Quant-dLLM consistently achieves higher accuracy than state-of-the-art (SOTA) AR-transfer PTQ methods on dLLMs. The code and models will be available at https://github.com/ZTA2785/Quant-dLLM

## 1 Introduction

Transformer-based large language models (LLMs) (Vaswani, 2017) have achieved strong results across a wide range of natural language processing and text generation tasks. Recent advances are led by autoregressive systems such as LLaMA (Touvron et al., 2023) and Qwen (Yang et al., 2025), driven by scaling model size to billions of parameters. Recently, diffusion-based large language models (dLLMs) have emerged as a promising alternative (Nie et al., 2025; Ye et al., 2025). They generate text by iteratively denoising masked sequences with bidirectional context and a timestep-dependent visibility schedule, which supports parallel token recovery and fine-grained control of structure. Efficient dLLM deployment remains difficult as multi-step denoising and large parameter counts increase memory. Thus, training-free weight compression is essential.

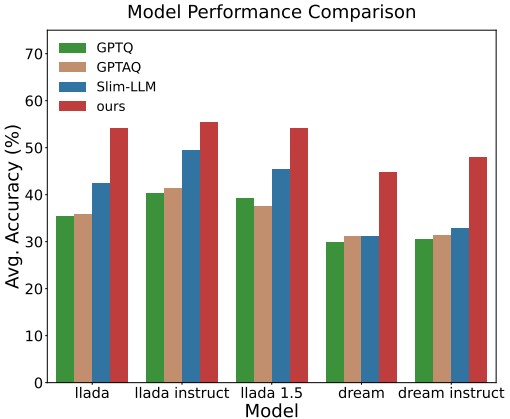

Figure 1: dLLMs' performance on 7 general tasks. Our Quant-dLLM yields the best accuracy at equal memory cost.

In recent years, a large body of work explores compression for autoregressive large language models, including weight quantization (Frantar et al., 2023; Lin et al., 2024; Ashkboos et al., 2024), low rank factorization (Zhang et al., 2024; Yuan et al., 2023), network pruning (Sun et al., 2024; Frantar & Alistarh, 2023), and knowledge distillation (Zhong et al., 2024; Gu et al., 2024). Among these

---

*Equal contribution
†Corresponding author: Yulun Zhang, yulun100@gmail.com

approaches, post-training quantization (PTQ) is especially attractive because it avoids finetuning and scales to billion-parameter models. On autoregressive models, PTQ has advanced with curvature-aware error control and with layout or rotation techniques that reduce outliers, and it delivers strong accuracy at low-bit precision with simple deployment. Recently, Lin et al. (2025) systematically transfers classic low-bit PTQ methods from autoregressive LLMs to diffusion language models and reports that weight-only 4-bit performs well, whereas 3-bit shows noticeable degradation on math and code tasks. Building on these observations, we extend weight-only PTQ to a 2-bit budget and find that performance drops sharply, which highlights the mismatch with diffusion-style inference.

This work focuses on 2-bit weight-only PTQ for dLLMs. Inspired by DB-LLM (Chen et al., 2024), we encode each weight matrix as a combination of binarized matrices with row–column scaling. This keeps bitwise-friendly execution and increases expressiveness under the same 2-bit budget. In dLLMs, we observe that it has lower reconstruction error than fixed 2-bit codes and more stable fitting under masked, timestep-dependent activations. The parameterization also reveals structured sparsity that reduces memory usage, making it a natural foundation for our diffusion-aligned PTQ.

In this work, we develop a training-free, weight-only PTQ pipeline tailored to dLLMs under a 2-bit budget. Our analysis finds two main sources of error at 2 bits. **(i)** Timestep and mask schedules shift the activation distribution away from fully visible calibration. **(ii)** Quantization errors accumulate across denoising steps and become larger at later stages. To address these issues, we first propose a diffusion-aligned calibration simulation that takes the inference visibility schedule: calibration batches are drawn with explicit coverage over timesteps and masking ratios. Building on these calibrated statistics, we develop a data-aware any-order quantizer that reconstructs each weight matrix as a combination of binarized matrices with row-column scaling. It is optimized with a Data-aware Objective Reformulation and Row-column Successive Re-scaling. The order $K$ can be increased while preserving binary-friendly execution (§3.3). Finally, under a strict 2-bit weight budget, we introduce adaptive blockwise mixed precision that assigns different orders to blocks by importance, granting higher effective precision to a small set of sensitive regions while keeping the majority 2-bit. This stabilizes late denoising steps without changing activation precision(§3.4).

Our key contributions can be summarized as follows:

- **Masked Calibration Simulation (MCS).** We construct timestep-aware, partially visible calibration inputs that mirror the diffusion denoising process, reducing the distribution mismatch between calibration and inference.

- **Data-aware Any-order Quantizer (DAQ).** We introduce a row-column scaled multi-binary parameterization with Data-aware Objective Reformulation (DOR) and Row-column Successive Re-scaling (RSR), and extend it to an any-order composition.

- **Adaptive Blockwise Mixed Precision (ABMP).** We propose an importance-guided, per-block order assignment that adheres to a strict 2-bit average budget, concentrating representational capacity on the most critical model components.

- **Quant-dLLM.** We present Quant-dLLM, which seamlessly integrates MCS, DAQ, and ABMP into standard PTQ pipelines, achieving state-of-the-art (SOTA) 2-bit weight-only accuracy on recent dLLMs. We will release the code soon.

## 2 RELATED WORKS

### 2.1 DIFFUSION LANGUAGE MODELS

Diffusion models have seen remarkable success in generating continuous data such as images, video, and audio by learning to reverse a forward noise process. However, extending this framework to language introduces a unique set of challenges due to the inherently discrete and structured nature of textual data. Early work addressed this by modeling the reverse dynamics of a discrete diffusion process with an absorbing state (He et al., 2022; Austin et al., 2021a). More recently, Masked Diffusion Models (MDMs) (Lou et al., 2023; Ou et al., 2024; Shi et al., 2024) have gained increasing attention by adopting a simpler forward process that gradually replaces input tokens with a designated [MASK] token. This year, efforts have successfully scaled these models to the billion-parameter regime. Notable examples include LLaDA (Nie et al., 2025), which uses a bidirectional transformer as a denoiser to achieve performance comparable to autoregressive LLMs, and Dream (Ye et al., 2025), which is initialized from a pre-trained autoregressive model and demonstrates competitive generation capabilities. These advancements confirm the viability of diffusion-based approaches as an alternative to autoregressive models for large-scale language modeling.

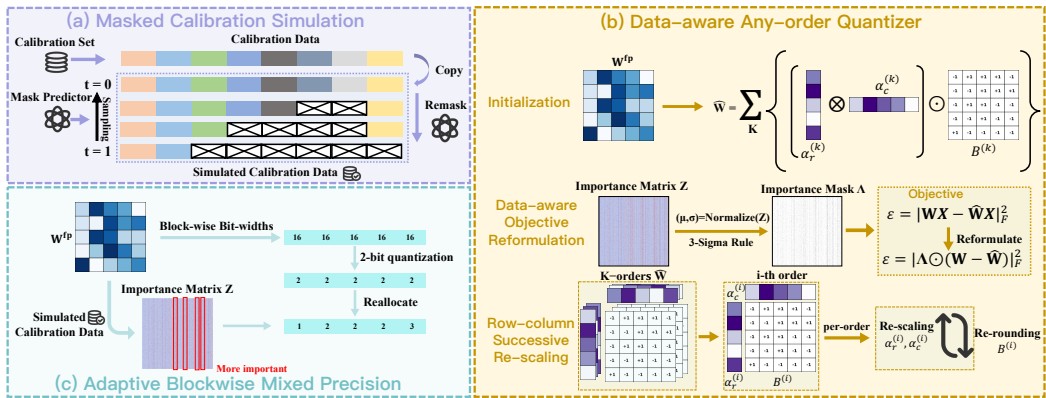

Figure 2: Overview of our Quant-dLLM. **Masked Calibration Simulation**: Aligns calibration with diffusion by simulating masked, timestep-aware inputs. **Adaptive Blockwise Mixed Precision**: Assigns binary orders by importance under a 2-bit average. **Data-aware Any-order Quantizer**: Builds multi-binary RC forms with data-aware optimization.

Despite these encouraging results, the deployment of dLLMs remains constrained by the computational demands of their billions of parameters. While some recent works propose caching strategies to accelerate dLLM inference (Wu et al., 2025; Liu et al., 2025; Ma et al., 2025), our research takes a different, complementary approach. We explore the potential PTQ techniques to dLLMs, aiming to reduce quantized model memory footprint while preserving generation quality.

### 2.2 WEIGHT-ONLY QUANTIZATION

Weight-only quantization is a primary strategy for compressing LLMs, typically categorized into quantization-aware training (QAT) and post-training quantization (PTQ). While QAT integrates quantization into the training process to achieve low-bit representations, it often demands significant computational resources and extensive training time. In contrast, PTQ applies quantization directly to a pre-trained model's weights, making it a faster and more resource-efficient alternative.

Recent advancements in PTQ for autoregressive LLMs include methods like GPTQ (Frantar et al., 2023) and GPTAQ that leverage Hessian-guided error compensation, as well as approaches such as Slim-LLM (Huang et al., 2024b), which employs mixed-precision schemes to enhance the perception of important weight information. Other techniques like AWQ Lin et al. (2024) use activation-aware scaling to achieve strong performance at low bitwidths. For extreme compression, binarization methods like BiLLM (Huang et al., 2024a) and ARB-LLM (Li et al., 2025b) push the boundaries to 1-bit, demonstrating competitive results. Sub-1-bit methods (Dong et al., 2025; Yan et al., 2025) push compression further by lowering average bitwidths while maintaining strong accuracy. A recent study by Lin et al. (2025) explored transferring these PTQ methods to dLLMs. While effective at 3-4 bits, we found that performance degraded severely at 2-bit precision, highlighting a notable challenge in ultra-low-bit quantization for dLLMs. To address this gap, we develop a tailored 2-bit weight-only PTQ framework, achieving significant improvement over the SOTA method for autoregressive LLMs.

## 3 METHOD

**Overview.** Figure 2 summarizes our pipeline. Section 3.1 reviews masked diffusion for language models and binarization. Section 3.2 introduces Masked Calibration Simulation (MCS), aligning calibration with timestep-dependent masking. Section 3.3 presents the Data-aware Any-order Quantizer (DAQ), which fits RC-scaled binary matrices via data-aware closed-form successive ee-scaling. Section 3.4 introduces Adaptive Blockwise Mixed Precision (ABMP), reallocating precision by importance, upgrading key blocks to 3-bit and reducing others to 1-bit under a strict 2-bit budget.

### 3.1 PRELIMINARY

**Masked diffusion for language models.** We consider a diffusion decoder that operates exclusively on masked token positions. For a token sequence $y \in \{e_1, \dots, e_K\}^L$, an absorbing token $m$ (i.e., [MASK]), and a monotonic visibility schedule $\alpha_t \in [0, 1]$, the forward noising process is defined as:

$$q(y_t \mid y) = \text{Cat}(\alpha_t y + (1 - \alpha_t) m). \tag{1}$$

The decoding (denoising) process proceeds from $t=1 \rightarrow 0$, where visible tokens are copied and masked tokens are predicted via $\boldsymbol{\pi}_\theta(y_t) = \text{softmax}(f_\theta(y_t))$.

**Binarization.** Given a weight matrix $\mathbf{W} \in \mathbb{R}^{n \times m}$, we row-centered matrix $\widetilde{\mathbf{W}} = \mathbf{W} - \mu$ with $\mu = \frac{1}{m} \sum_{j=1}^{m} \mathbf{W}_{:,j}$ and approximate it by a binary carrier $\mathbf{B} \in \{-1, +1\}^{n \times m}$ modulated by separable row-column scales: $\hat{\mathbf{W}} = (\boldsymbol{\alpha}_r \, \boldsymbol{\alpha}_c^\top) \odot \mathbf{B}, \boldsymbol{\alpha}_r \in \mathbb{R}^n, \; \boldsymbol{\alpha}_c \in \mathbb{R}^m$. The scales and carrier are obtained by

$$\min_{\boldsymbol{\alpha}_r, \boldsymbol{\alpha}_c, \mathbf{B}} \left\| \widetilde{\mathbf{W}} - (\boldsymbol{\alpha}_r \, \boldsymbol{\alpha}_c^\top) \odot \mathbf{B} \right\|_F^2, \tag{2}$$

This is typically solved via closed-form alternating updates for $\boldsymbol{\alpha}_r, \boldsymbol{\alpha}_c$ and sign rounding for $\mathbf{B}$ (see §3.3). The single-scale case recovers the classic solution $\mathbf{B} = \text{sign}(\widetilde{\mathbf{W}})$, $(\alpha_r)_i = \frac{1}{m} \sum_j |\widetilde{\mathbf{W}}_{ij}|$.

---

**Algorithm 1** MCS: Masked Calibration Simulation

---

func MCS($\mathcal{X}, T, \gamma, \alpha(t)$)
**Input:** $\mathcal{X}$ — full sequences; $T$ — total timesteps; $\gamma$ — prefix ratio; $\alpha(t)$ — visibility schedule
**Output:** Masked calibration set $\mathcal{D}_{\text{calib}}$
 1: $\mathcal{D}_{\text{calib}} \leftarrow \emptyset$
 2: **for all** $x \in \mathcal{X}$ **do**
 3:    $L \leftarrow$ length of $x$
 4:    $\mathcal{P} \leftarrow \{1, \ldots, \lfloor \gamma \cdot L \rfloor\}$
 5:    **for** $t = 1$ to $T$ **do**
 6:       $\alpha \leftarrow \alpha(t)$
 7:       **for** $i = 1$ to $L$ **do**
 8:          $r_i \leftarrow 1$ if $i \in \mathcal{P}$ else Bernoulli($\alpha$)
 9:          $\tilde{x}_i(t) \leftarrow x_i$ if $r_i = 1$ else [MASK]
10:       **end for**
11:       $\mathcal{D}_{\text{calib}} \leftarrow \mathcal{D}_{\text{calib}} \cup \{\tilde{x}(t)\}$
12:    **end for**
13: **end for**
14: **return** $\mathcal{D}_{\text{calib}}$

---

## 3.2 MASKED CALIBRATION SIMULATION (MCS)

Conventional calibration techniques assume autoregressive activations and static token visibility. However, such assumptions fail to capture the distributional shift introduced by timestep-dependent masking in dLLMs, leading to inaccurate quantization statistics and performance degradation. To address this, we propose Masked Calibration Simulation (MCS), a method that aligns the calibration process with the model's native denoising mechanism, as detailed in Algorithm 1.

MCS simulates activation distributions by constructing timestep-aware batches that jointly stratify across time and visibility ratio. This simulates the temporal uncertainty and partial token observations encountered during inference. Given a fully visible sequence $x \in \mathcal{V}^L$, we first fix a deterministic visible prefix: $\mathcal{P} = \{1, \ldots, \lfloor \gamma \cdot L \rfloor\}$. For each timestep $t \sim \pi(t)$, we first compute the corresponding visibility ratio $\alpha(t)$, which determines the expected proportion of visible tokens. Then, we construct a binary mask $r \in \{0, 1\}^L$ such that tokens in the always-visible set $\mathcal{P}$ are assigned $r_i = 1$, while the remaining positions are independently sampled from a Bernoulli distribution with probability $\alpha(t)$, i.e., $r_i = z_i \sim \text{Bernoulli}(\alpha(t))$ for $i \notin \mathcal{P}$. Finally, we set the masked input as $\tilde{x}_i(t) = x_i$ if $r_i = 1$, and $\tilde{x}_i(t) = $ [MASK] otherwise.

Applying this over a grid of $T$ timesteps yields a calibration dataset whose activation statistics closely match masked denoising. We then aggregate second-order statistics over this set. In §3.3, we further convert these into an elementwise importance mask to weight positions adaptively. For implementation, we discretize $[0, 1]$ into $T$ uniform timesteps, use a default $\gamma = 0.25$, and fix random seeds to ensure full experimental reproducibility.

## 3.3 DATA-AWARE ANY-ORDER QUANTIZER (DAQ)

To effectively parameterize weights under the masked activation statistics from MCS, we introduce the Data-aware Any-order Quantizer (DAQ). Instead of mapping weights to a small set of fixed quantization levels, DAQ approximates each weight matrix as a combination of several binary matrices, each modulated by its own separable row-column scaling factors $(\boldsymbol{\alpha}_r \, \boldsymbol{\alpha}_c^\top)$. This formulation, expressed as a superposition of binarized components, preserves the computational efficiency of binary operations while significantly expanding the model's representational capacity:

$$\hat{\mathbf{W}} = \sum_{k=1}^{K} \left( \boldsymbol{\alpha}_r^{(k)} \, \boldsymbol{\alpha}_c^{(k)\top} \right) \odot \mathbf{B}_k, \qquad \mathbf{B}_k \in \{-1, +1\}^{n \times m}. \tag{3}$$

The integer $K$ denotes the order of the representation, which directly corresponds to the bit-width assigned to a weight block (§3.4).

We begin by analyzing the first-order case ($K = 1$), where a single row-column scale pair modulates one binary matrix: $\hat{\mathbf{W}} = (\boldsymbol{\alpha}_r \, \boldsymbol{\alpha}_c^\top) \odot \mathbf{B}$. While minimizing weight reconstruction error is a common proxy, a more direct objective is to minimize the error on the layer's output, thereby better aligning

the quantization process with the model's behavior during inference. To this end, we incorporate the simulated calibration data $\mathbf{X}$ from MCS and define the data-aware quantization error as: $\mathcal{L}_2 = ||\mathbf{W}\mathbf{X} - \widehat{\mathbf{W}}\mathbf{X}||_F^2$. However, directly computing this objective is computationally prohibitive due to the large number of matrix multiplications involving the calibration data $\mathbf{X}$. To create a tractable objective, we follow prior work (Li et al., 2025b) and reformulate the error. By pre-computing the uncentered second moment of the simulated activations:

$$\mathbf{S}_{\text{MCS}} = \mathbb{E}_{t \sim \pi}\Big[ \sum_b \tilde{\mathbf{X}}_b(t)\, \tilde{\mathbf{X}}_b(t)^\top \Big] \approx \frac{1}{|\mathcal{T}|} \sum_{t \in \mathcal{T}} \sum_b \tilde{\mathbf{X}}_b(t)\, \tilde{\mathbf{X}}_b(t)^\top, \tag{4}$$

the objective can be expressed as an efficient quadratic form: $\mathcal{L}_2 = \text{Tr}\big((\mathbf{W}-\widehat{\mathbf{W}})\, \mathbf{S}_{\text{MCS}}\, (\mathbf{W}-\widehat{\mathbf{W}})^\top\big)$. While this reformulation is efficient, directly optimizing the quadratic objective remains challenging. Therefore, we develop the following strategies to create a tractable yet effective approximation.

---

**Algorithm 2** DAQ: Data-aware any-order Quantizer

---

func DAQ($\mathbf{W}$, $\mathbf{Z}$, $K$, $T$, $\lambda$)
**Input** $\mathbf{W}, \mathbf{Z} \in \mathbb{R}^{n \times m}$, $K, T \in \mathbb{N}$, $\lambda > 1$
**Output**

$$\hat{\mathbf{W}} = \sum_{k=1}^{K} \big(\boldsymbol{\alpha}_r^{(k)} \boldsymbol{\alpha}_c^{(k)\top}\big) \odot \mathbf{B}^{(k)}$$

1: $\boldsymbol{\Lambda} := \text{build\_importance\_mask}(\mathbf{Z}, \lambda)$
2: $\hat{\mathbf{W}} := \mathbf{0}_{n \times m}$
3: **for** $k = 1, 2, \ldots, K$ **do**
4:     $\mathbf{R} \leftarrow \mathbf{W} - \hat{\mathbf{W}}$
5:     $\boldsymbol{\alpha}_r^{(k)}, \boldsymbol{\alpha}_c^{(k)}, \mathbf{B}^{(k)} \leftarrow \text{binary\_rc\_init}(\mathbf{R})$
6:     $\mathbf{S}^{(k)} \leftarrow \boldsymbol{\alpha}_r^{(k)}\big(\boldsymbol{\alpha}_c^{(k)}\big)^\top$
7:     $\hat{\mathbf{W}} \leftarrow \hat{\mathbf{W}} + \mathbf{S}^{(k)} \odot \mathbf{B}^{(k)}$
8: **end for**
9: **for** $t = 1, 2, \ldots, T$ **do**
10:     **for** $k = 1, 2, \ldots, K$ **do**
11:        $\hat{\mathbf{W}}_{\backslash k} \leftarrow \sum_{q \neq k} \big(\mathbf{S}^{(q)} \odot \mathbf{B}^{(q)}\big)$
12:        $\mathbf{R}^{(k)} \leftarrow \mathbf{W} - \hat{\mathbf{W}}_{\backslash k}$
13:        $\boldsymbol{\alpha}_r^{(k)} \leftarrow \text{update\_}\alpha_\text{r}(\mathbf{R}^{(k)}, \mathbf{B}^{(k)}, \boldsymbol{\alpha}_c^{(k)}, \boldsymbol{\Lambda})$
14:        $\boldsymbol{\alpha}_c^{(k)} \leftarrow \text{update\_}\alpha_\text{c}(\mathbf{R}^{(k)}, \mathbf{B}^{(k)}, \boldsymbol{\alpha}_r^{(k)}, \boldsymbol{\Lambda})$
15:        $\mathbf{S}^{(k)} \leftarrow \boldsymbol{\alpha}_r^{(k)}\big(\boldsymbol{\alpha}_c^{(k)}\big)^\top$
16:     **end for**
17:     $\{\mathbf{B}^{(k)}\}_{k=1}^K \leftarrow \text{update\_B}\Big(\mathbf{W}, \{\boldsymbol{\alpha}_r^{(k)}\}_{k=1}^K,$
                    $\{\boldsymbol{\alpha}_c^{(k)}\}_{k=1}^K\Big)$
18:     $\hat{\mathbf{W}} \leftarrow \sum_{k=1}^K \mathbf{S}^{(k)} \odot \mathbf{B}^{(k)}$
19: **end for**
20: **return** $\hat{\mathbf{W}}$

func binary\_rc\_init($\mathbf{X}$)
1: $\boldsymbol{\alpha}_r \leftarrow \frac{1}{m} |\mathbf{X}|\, \mathbf{1}_m$
2: $\boldsymbol{\alpha}_c \leftarrow \frac{1}{n} |\mathbf{X}|^\top \text{diag}(\boldsymbol{\alpha}_r)^{-1}\, \mathbf{1}_n$
3: $\mathbf{B} \leftarrow \text{sign}(\mathbf{X})$
4: **return** $\boldsymbol{\alpha}_r, \boldsymbol{\alpha}_c, \mathbf{B}$

func update\_$\alpha_\text{r}$($\mathbf{X}$,$\mathbf{B}$,$\boldsymbol{\alpha}_c$,$\boldsymbol{\Lambda}$)
1: $\mathbf{u} \leftarrow (\boldsymbol{\Lambda}^2 \odot \mathbf{X} \odot \mathbf{B})\, \boldsymbol{\alpha}_c$
2: $\mathbf{v} \leftarrow (\boldsymbol{\Lambda}^2\, (\text{diag}\,\boldsymbol{\alpha}_c)\, \boldsymbol{\alpha}_c) + \varepsilon\, \mathbf{1}_n$
3: **return** $\mathbf{u} \oslash \mathbf{v}$

func update\_$\alpha_\text{c}$($\mathbf{X}$,$\mathbf{B}$,$\boldsymbol{\alpha}_r$,$\boldsymbol{\Lambda}$)
1: $\mathbf{u} \leftarrow (\boldsymbol{\Lambda}^2 \odot \mathbf{X} \odot \mathbf{B})^\top \boldsymbol{\alpha}_r$
2: $\mathbf{v} \leftarrow (\boldsymbol{\Lambda}^2)^\top (\text{diag}(\boldsymbol{\alpha}_r)\, \boldsymbol{\alpha}_r) + \varepsilon\, \mathbf{1}_m$
3: **return** $\mathbf{u} \oslash \mathbf{v}$

func update\_B$\Big(\mathbf{W}, \{\boldsymbol{\alpha}_r^{(k)}\}_{k=1}^K, \{\boldsymbol{\alpha}_c^{(k)}\}_{k=1}^K\Big)$
1: **for** $i = 1, \ldots, n$ **do**
2:     **for** $j = 1, \ldots, m$ **do**
3:        $\mathbf{s} \leftarrow \big[(\alpha_r^{(k)})_i (\alpha_c^{(k)})_j\big]_{k=1}^K$
4:        $\mathbf{b} \leftarrow \text{search}(W_{ij}, \mathbf{s})$
5:        **for** $k = 1, \ldots, K$ **do**
6:           $(\mathbf{B}^{(k)})_{ij} \leftarrow \mathbf{b}_k$
7:        **end for**
8:     **end for**
9: **end for**
10: **return** $\{\mathbf{B}^{(k)}\}_{k=1}^K$

func build\_importance\_mask($\mathbf{Z}$, $\lambda$)
1: $\mu \leftarrow \text{mean}(\mathbf{Z}), \quad \sigma \leftarrow \text{std}(\mathbf{Z})$
2: $\mathbf{M} \leftarrow \mathbb{I}(|\mathbf{Z} - \mu| > 3\sigma)$
3: **return** $\mathbf{1}_{n \times m} + (\lambda - 1)\, \mathbf{M}$

---

**Data-aware Objective Reformulation (DOR).** Our analysis reveals that quantization error is not uniformly distributed but concentrates on a small, critical subset of weights. This insight allows us to distill the complex, data-dependent information from $\mathbf{S}_{\text{MCS}}$ into a computationally efficient proxy. We construct a importance matrix $\mathbf{Z} \in \mathbb{R}^{n \times m}$ with the same shape as $\mathbf{W}$. Let $\mathbf{d} = \text{diag}\big((\mathbf{S}_{\text{MCS}} + \gamma\, \mathbf{I}_m)^{-1}\big) \in \mathbb{R}^m$ and $\mathbf{D} = \text{diag}(\mathbf{d})$. We normalize $\mathbf{W}$ column-wise by $\mathbf{D}^{-1}$ and square elementwise: $\mathbf{Z} = (\mathbf{W}\mathbf{D}^{-1}) \odot (\mathbf{W}\mathbf{D}^{-1})$. We then standardize $\mathbf{Z}$ along a chosen block on the calibration set and detect outliers with the $3\sigma$ rule:

$$\widetilde{\mathbf{Z}} = (\mathbf{Z} - \boldsymbol{\mu}) \oslash \boldsymbol{\sigma}, \qquad \boldsymbol{\Pi} = \mathbb{I}(|\widetilde{\mathbf{Z}}| > 3). \tag{5}$$

The $3\sigma$ threshold isolates the small heavy-tailed portion of high-importance weights that drives most of the error, while keeping the selected fraction compact and stable across layers. Finally, we form the importance mask: $\boldsymbol{\Lambda} = \mathbf{1}_{n \times m} + (\lambda - 1)\, \boldsymbol{\Pi}$, which means elements beyond $3\sigma$ receive weight $\lambda > 1$ and the others receive weight 1. Therefore, we distill the effect of $\mathbf{S}_{\text{MCS}}$ into $\boldsymbol{\Lambda}$ and optimize

the data-aware Frobenius proxy:

$$\widehat{\mathcal{L}}_{\mathbf{\Lambda}}(\boldsymbol{\alpha}_r, \boldsymbol{\alpha}_c) \;=\; \left\| \mathbf{\Lambda} \odot (\mathbf{W} - \hat{\mathbf{W}}) \right\|_F^2 \;=\; \mathrm{Tr}\Big( (\mathbf{\Lambda} \odot (\mathbf{W} - \hat{\mathbf{W}})) (\mathbf{\Lambda} \odot (\mathbf{W} - \hat{\mathbf{W}}))^{\top} \Big). \quad (6)$$

When $\mathbf{\Lambda} = \mathbf{1}_{n \times m}$, this objective reduces to the standard Frobenius reconstruction error.

**Row-column Successive Re-scaling (RSR).** We minimize the data-aware Frobenius proxy using an iterative method RSR. The optimization objective is given by:

$$\widehat{\mathcal{L}}_{\mathbf{\Lambda}}(\boldsymbol{\alpha}_r, \boldsymbol{\alpha}_c) = \left\| \mathbf{\Lambda} \odot \big(\mathbf{W} - (\boldsymbol{\alpha}_r \boldsymbol{\alpha}_c^{\top}) \odot \mathbf{B}\big) \right\|_F^2. \quad (7)$$

The RSR algorithm operates by alternately optimizing one set of scaling factors while holding the other fixed. By differentiating $\widehat{\mathcal{L}}_{\mathbf{\Lambda}}$ with respect to each set of scales and setting the gradient to zero, we obtain closed-form update rules that guarantee a monotonic decrease in the objective:

$$\boldsymbol{\alpha}_r = \big[ (\mathbf{\Lambda^2} \odot \mathbf{W} \odot \mathbf{B}) \, \boldsymbol{\alpha}_c \big] \; \oslash \; \big[ (\mathbf{\Lambda^2} (\mathrm{diag}(\boldsymbol{\alpha}_c) \, \boldsymbol{\alpha}_c) + \varepsilon \, \mathbf{1}_n \big], \quad (8)$$

$$\boldsymbol{\alpha}_c = \big[ (\mathbf{\Lambda^2} \odot \mathbf{W} \odot \mathbf{B})^{\top} \boldsymbol{\alpha}_r \big] \; \oslash \; \big[ (\mathbf{\Lambda^2} (\mathrm{diag}(\boldsymbol{\alpha}_r) \, \boldsymbol{\alpha}_r) + \varepsilon \, \mathbf{1}_m \big]. \quad (9)$$

Following the updates to the scaling factors, the binary matrix $\mathbf{B}$ is refined through a per-entry minimal-error search:

$$B_{ij} \; = \arg \min_{b \in \{\pm 1\}} \; \big| W_{ij} - \big(\alpha_{r,i} \alpha_{c,j}\big) b \big|. \quad (10)$$

Here, $\odot$ and $\oslash$ denote element-wise product and division, respectively. $\mathrm{diag}(\cdot)$ forms a diagonal matrix, $\mathbf{1}_n, \mathbf{1}_m$ are all-ones vectors, and $\varepsilon > 0$ is a small constant to stabilize division.

**K-order extension.** The K-order representation is constructed through a greedy, iterative process that extends the single-order framework. We begin by finding the optimal first-order approximation $\hat{\mathbf{W}}^{(1)}$. Then, we compute the residual $\mathbf{R}^{(1)} = \mathbf{W} - \hat{\mathbf{W}}^{(1)}$ and apply the same RSR procedure to find the optimal approximation for this residual. This process is repeated $K$ times, with each new binary component fitting the residual from the previous stage. This formulation naturally extends to any order $K$, progressively optimizing the approximation. The full procedure is detailed in Algorithm 2.

---

**Algorithm 3** ABMP: Adaptive Blockwise Mixed Precision

1: Given block partition $\mathcal{G}$ and importance $\mathbf{Z}$
2: Compute $s_g = \sum_{(i,j) \in g} \mathbf{Z}_{ij}$ for all $g \in \mathcal{G}$
3: Set $k \leftarrow \lfloor 0.05 |\mathcal{G}| \rfloor$ and sort blocks by $s_g$
4: Assign $b_g = 3$ to top-$k$, $b_g = 1$ to bottom-$k$, others $b_g = 2$   ($\frac{1}{|\mathcal{G}|} \sum_g b_g = 2$)
5: Apply DAQ quantization to each block using its assigned order $K = b_g$.

---

## 3.4 ADAPTIVE BLOCKWISE MIXED PRECISION (ABMP)

The DAQ representation from §3.3 provides a flexible multi-binary parameterization. However, a uniform application of order $K$ is inherently suboptimal. Such a strategy fails to differentiate between model components of varying importance, leading to a misallocation of the bit budget, which means highly salient regions suffer from performance degradation due to insufficient precision, while less critical regions consume more bits than necessary.

To address this, we introduce Adaptive Blockwise Mixed Precision (ABMP), a strategy that intelligently redistributes the quantization budget at a block level. The core principle of ABMP is to strategically reallocate precision under a strict per-layer budget, ensuring the model's average bit-width remains exactly at 2-bit:

$$\frac{1}{|\mathcal{G}|} \sum_{g \in \mathcal{G}} b_g \;=\; 2, \qquad b_g \in \{1, 2, 3\}. \quad (11)$$

This approach concentrates representational capacity where it is most impactful, enhancing model fidelity without increasing the overall memory footprint. The allocation process is guided by the importance of each block. To quantify this importance, we leverage the elementwise importance matrix $\mathbf{Z}$ (from §3.3) to compute an aggregate importance score $s_g = \sum_{(i,j) \in g} \mathbf{Z}_{ij}$ for each block $g \in \mathcal{G}$. We then rank all blocks within a layer based on these scores. To enforce the 2-bit average budget, the number of blocks assigned a higher precision (3-bit) is precisely matched by an equal number of blocks assigned a lower precision (1-bit). Specifically, we allocate 3-bit precision to the top-$k$ most important blocks and 1-bit precision to the bottom-$k$.

Based on our experimental analysis, we set the default reallocation ratio to 5% per layer (i.e., $k = \lfloor 0.05 |\mathcal{G}| \rfloor$). As detailed in our ablation studies, this ratio provides a robust balance between performance and stability, with diminishing returns observed beyond 10%. The complete procedure is outlined in Algorithm 3. By dynamically adapting the bit-width at a structured, block-wise level, ABMP effectively preserves critical information in dLLMs, which is particularly crucial for stabilizing performance during the later, more sensitive stages of the denoising process.

### 3.5 QUANT-DLLM PIPELINE

As illustrated in Figure 2, our proposed Quant-dLLM framework is composed of three key components: Masked Calibration Simulation (MCS), Data-aware Any-order Quantizer (DAQ), and Adaptive Blockwise Mixed Precision (ABMP). The end-to-end pipeline operates in a layer-wise manner, requiring only a pre-trained dLLM and a small set of calibration data.

The workflow begins by first processing the calibration data with MCS (§3.2) to generate a timestep-aware dataset whose activation statistics align with the diffusion-denoising process. Subsequently, the framework proceeds layer by layer. For each target layer, we first compute an importance matrix $\mathbf{Z}$ from the simulated calibration data. This matrix is then passed to ABMP (§3.4), which determines the optimal per-block bit-width assignment $\{b_g\}$ under a strict 2-bit average budget. Finally, DAQ (§3.3) quantizes the layer's weight matrix according to the bit-widths allocated by ABMP. This process is repeated for all targeted layers until the entire model is quantized. Since both quantization and bit allocation are performed layer-wise, our framework maintains a low memory overhead and avoids any gradient-based updates, making it a highly efficient PTQ solution.

## 4 EXPERIMENTS

### 4.1 EXPERIMENTAL SETTINGS

All experiments are conducted on a single NVIDIA A800-80GB GPU using the PyTorch and Huggingface frameworks. As a post-training quantization (PTQ) framework, Quant-dLLM does not require any training or gradient backpropagation. We use 128 calibration samples from the C4 dataset, with each sequence having a length of 4096 tokens, and the group size is fixed at 128 for all experiments. For evaluation, we use recent dLLMs, including LLaDA-8B-Base (Nie et al., 2025), LLaDA-8B-Instruct (Nie et al., 2025), LLaDA-1.5 (Zhu et al., 2025), Dream-7B-Base Ye et al. (2025), and Dream-7B-Instruct (Ye et al., 2025). We evaluate Quant-dLLM on three task categories:

- **General-knowledge QA:** MMLU (Hendrycks et al., 2020), ARC-E/C (Clark et al., 2018), HellaSwag (Zellers et al., 2019), WinoGrande (Sakaguchi et al., 2020), PIQA (Bisk et al., 2020), and BBH (Suzgun et al., 2022).
- **Mathematical and scientific reasoning:** GSM8K (Cobbe et al., 2021), MATH (Hendrycks et al., 2021), and GPQA (Rein et al., 2024).
- **Code generation:** HumanEval (Chen et al., 2021) and MBPP (Austin et al., 2021b).

General tasks are evaluated in an n-shot setting with 128 Monte Carlo samples, while code tasks report the Pass@1 metric. We compare our method against three strong PTQ baselines tailored for low-bit regimes: GPTQ (Frantar et al., 2023), a traditional block-wise weight-only quantization method; GPTAQ (Li et al., 2025a), a variant of GPTQ that incorporates activation into its compensation; and Slim-LLM (Huang et al., 2024b), a mixed-precision weight-only SOTA PTQ method. Baselines use the same calibration data and sequence length as ours.

### 4.2 MAIN RESULTS

Table 1 reports the weight-only quantization results for GPTQ, GPTAQ, Slim-LLM, and our Quant-dLLM on five different dLLM models across seven general knowledge tasks. Our method consistently delivers the highest average accuracy among 2-bit baselines on all five models, surpassing GPTQ, GPTAQ, and Slim-LLM by a clear margin. Averaged over all models, our method improves the mean score from 36.5 (GPTQ), 35.6 (GPTAQ), and 40.9 (Slim-LLM) to 51.3.

On a per-model basis, our approach consistently improves the average score over Slim-LLM. For LLaDA-8B-Base, the average score rises from 42.39 to **54.06**, an improvement of over 27%. Our method also preserves a large fraction of the full-precision performance on the LLaDA series under the same extreme 2-bit constraints. For example, on LLaDA-8B-Base, our average score reaches 87.72% of the full-precision score. Beyond the average, our method provides consistent per-task improvements over Slim-LLM and achieves the best score on all seven general benchmarks for all

Table 1: Results of GPTQ, GPTAQ, Slim-LLM, and our Quant-dLLM with 2-bit weight quantization among 7 tasks on LLaDA-Base, LLaDA-Instruct, LLaDA-1.5, Dream-Base, and Dream-Instruct. The numbers in parentheses represent the number used for evaluation. Best results are marked in **bold**.

| Model | Method | MMLU(5) | Wino.(5) | PIQA(0) | ARC-C(0) | ARC-E(0) | Hella.(0) | BBH(0) | Avg |
|---|---|---|---|---|---|---|---|---|---|
| **LLaDA-8B-Base** | FP | 65.76 | 74.59 | 74.70 | 44.11 | 74.20 | 54.27 | 42.60 | 61.46 |
| | GPTQ | 34.75 | 57.85 | 57.45 | 22.44 | 37.04 | 33.77 | 4.06 | 35.34 |
| | GPTAQ | 34.73 | 59.04 | 56.42 | 22.27 | 38.17 | 35.13 | 5.34 | 35.87 |
| | Slim-LLM | 47.98 | 60.85 | 62.46 | 26.11 | 53.41 | 39.25 | 6.64 | 42.39 |
| | **Quant-dLLM** | **56.87** | **68.19** | **69.75** | **36.26** | **68.69** | **46.45** | **32.18** | **54.06** |
| **LLaDA-8B-Instruct** | FP | 63.91 | 72.30 | 74.43 | 53.24 | 79.34 | 53.95 | 47.89 | 63.58 |
| | GPTQ | 42.15 | 58.88 | 61.21 | 27.22 | 51.18 | 37.46 | 3.93 | 40.29 |
| | GPTAQ | 44.94 | 60.22 | 61.04 | 29.01 | 52.44 | 37.18 | 4.55 | 41.34 |
| | Slim-LLM | 50.35 | 60.69 | 65.72 | 34.73 | 65.57 | 41.34 | 27.48 | 49.41 |
| | **Quant-dLLM** | **54.07** | **65.67** | **70.73** | **43.86** | **71.97** | **47.39** | **35.02** | **55.53** |
| **LLaDA-1.5** | FP | 64.21 | 54.38 | 74.54 | 54.27 | 79.80 | 54.38 | 56.27 | 62.55 |
| | GPTQ | 48.48 | 37.47 | 61.48 | 30.20 | 52.90 | 37.47 | 6.91 | 39.27 |
| | GPTAQ | 47.26 | 35.89 | 58.81 | 28.41 | 51.39 | 35.89 | 5.38 | 37.58 |
| | Slim-LLM | 44.99 | 41.86 | 65.89 | 33.70 | 64.86 | 41.86 | 24.90 | 45.44 |
| | **Quant-dLLM** | **54.32** | **47.82** | **71.16** | **47.18** | **73.06** | **47.82** | **38.09** | **54.21** |
| **Dream-7B-Base** | FP | 69.43 | 73.56 | 74.59 | 55.38 | 82.58 | 54.42 | 50.12 | 65.73 |
| | GPTQ | 23.84 | 50.51 | 53.86 | 19.88 | 28.70 | 29.69 | 3.08 | 29.94 |
| | GPTAQ | 23.90 | 52.49 | 55.88 | 20.82 | 30.43 | 30.98 | 4.00 | 31.21 |
| | Slim-LLM | 25.54 | 51.62 | 54.24 | 20.05 | 31.14 | 30.83 | 4.18 | 31.09 |
| | **Quant-dLLM** | **40.22** | **59.19** | **63.71** | **29.35** | **54.29** | **40.87** | **25.59** | **44.75** |
| **Dream-7B-Instruct** | FP | 69.71 | 72.93 | 75.14 | 57.25 | 83.80 | 54.43 | 57.92 | 67.31 |
| | GPTQ | 23.96 | 50.59 | 54.90 | 18.52 | 30.77 | 30.11 | 5.32 | 30.60 |
| | GPTAQ | 24.32 | 49.72 | 56.42 | 20.73 | 31.82 | 30.97 | 5.11 | 31.30 |
| | Slim-LLM | 25.29 | 51.93 | 56.09 | 20.56 | 36.99 | 31.55 | 7.61 | 32.86 |
| | **Quant-dLLM** | **43.14** | **57.93** | **66.59** | **33.62** | **62.96** | **41.36** | **30.35** | **47.99** |

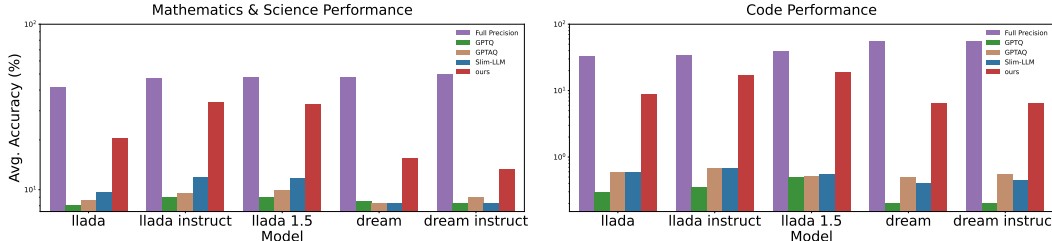

Figure 3: Average accuracy of mathematical & scientific reasoning, and code generation datasets on LLaDA series and Dream series.

five models tested. These results demonstrate that under strict 2-bit weight quantization, our approach delivers robust and transferable accuracy improvements across different dLLM families.

In Figure 3, we further illustrate our method's superiority on mathematics, science, and code tasks. Previous methods show a significant performance collapse, while our Quant-dLLM is the only method that effectively retains accuracy, providing a robust solution for a wide range of applications. For example, on the LLaDA-Instruct model, our method achieves an average accuracy of over 30% for mathematics and science tasks, while all baselines remain below 12%. For code generation, our method achieves over 15% accuracy on LLaDA-Instruct and LLaDA-1.5, while baselines hover near 0%. This remarkable resilience highlights that our Quant-dLLM pipeline achieves a significant performance increase compared to prior methods.

## 4.3 ABLATION STUDY

Unless otherwise specified, all ablation studies are conducted on LLaDA-8B-Base and Dream-7B-Base, reporting the 5-shot accuracy on MMLU under a strict 2-bit average budget, where group size and block size are both 128. We vary one factor at a time while keeping all other settings fixed.

**Effectiveness of MCS.** To verify the effectiveness of our Masked Calibration Simulation (MCS), we compare the performance of Quant-dLLM with and without MCS. As shown in Table 2a, on the LLaDA-8B-Base model, the MMLU accuracy is improved from 52.10% to 56.87% when using MCS. On the Dream-7B-Base model, the MMLU accuracy improves from 37.81% to 40.22%. These results demonstrate that aligning the calibration data with the denoising process of the diffusion model, MCS effectively reduces distribution mismatch, thereby significantly improving quantization performance.

Table 2: Ablation studies on LLaDA-8B-Base and Dream-7B-Base. We report MMLU in 5 shots.

(a) Effectiveness of MCS

| Model | MCS | MMLU(5) ↑ |
|---|---|---|
| LLaDA-8B-Base | ✗ | 52.10 |
| | ✓ | 56.87 |
| Dream-7B-Base | ✗ | 37.81 |
| | ✓ | 40.22 |

(b) Effectiveness of ABMP

| Model | ABMP | 0% | 5% | 10% | 15% |
|---|---|---|---|---|---|
| LLaDA-8B-Base | ✗ | 54.32 | – | – | – |
| | ✓ | – | 56.87 | 55.87 | 53.01 |
| Dream-7B-Base | ✗ | 32.75 | – | – | – |
| | ✓ | – | 34.50 | 40.22 | 31.11 |

(c) Ablation Study for DAQ

| Model | DAQ Component | MMLU(5) ↑ |
|---|---|---|
| LLaDA-8B-Base | baseline | 39.26 |
| | RSR w/o DOR | 48.32 |
| | RSR w/ DOR | 56.87 |
| Dream-7B-Base | baseline | 27.84 |
| | RSR w/o DOR | 34.73 |
| | RSR w/ DOR | 40.22 |

(d) Ablation Study for Calibration Set Size.

| Model | Calib. Data Size | MMLU(5) ↑ |
|---|---|---|
| LLaDA-8B-Base | 64 | 54.49 |
| | 128 | 56.87 |
| | 256 | 55.59 |
| Dream-7B-Base | 64 | 38.24 |
| | 128 | 40.22 |
| | 256 | 39.64 |

**Effectiveness of ABMP.** We investigate the impact of enabling ABMP and varying its bit allocation ratio. As shown in Table 2b, simply enabling ABMP provides a significant improvement over the uniform 2-bit baseline. On LLaDA-8B-Base, a 5% reallocation ratio achieves the best MMLU accuracy of 56.87%, surpassing the 54.32% baseline. On Dream-7B-Base, a 10% ratio performs best, increasing accuracy from 32.75% to 40.22%. This confirms that strategically reallocating precision to the most critical regions is a highly effective strategy for boosting overall performance.

**Ablation Study for DAQ.** We conduct a breakdown study to evaluate the contributions of the core components within our DAQ, as shown in Table 2c. The baseline represents a simple quantizer without our proposed enhancements. By introducing RSR without the data-aware objective, performance substantially increases, with MMLU accuracy rising from 39.26% to 48.32% on LLaDA-8B-Base. Incorporating the full DAQ with the data-aware objective (RSR w/ DOR) provides another major boost to 56.87%. A similar trend is observed on Dream-7B-Base. This clearly indicates that both the iterative re-scaling and the data-aware objective are crucial for DAQ's state-of-the-art performance.

**Ablation Study for Calibration Set Size.** We evaluate the impact of the calibration set size on Quant-dLLM's performance. As shown in Table 2d, using 128 samples, each with a sequence length of 4,096, yields the best performance. On LLaDA-8B-Base, increasing from 64 to 128 samples improves the MMLU accuracy from 54.49% to 56.87%. A further increase to 256 samples leads to a slight drop to 55.59%. Once a moderate threshold is met, using more samples doesn't significantly improve performance. This shows Quant-dLLM is practical for resource-limited environments.

## 4.4 MODEL SIZE ANALYSIS

Efficient model size reduction is critical for deploying dLLMs on resource-constrained hardware. We evaluate LLaDA-8B-Base under different quantization schemes and observe that Quant-dLLM achieves the smallest model size while maintaining accuracy. This advantage comes from our row–column scaling factor design, which imposes less memory overhead than the conventional zero-point and scaling

Table 3: Model size of LLaDA-8B-Base under different methods.

| Model | Method | Bit | Memory (GB) |
|---|---|---|---|
| LLaDA-8B | FP16 | 16 | 16.09 |
| | GPTQ | 2 | 3.70 |
| | Slim-LLM | 2 | 3.72 |
| | Quant-dLLM | 2 | 3.69 |

factor used in typical low-bit quantization. The detailed comparison is summarized in Table 3.

## 5 CONCLUSION

In this paper, we propose Quant-dLLM, a novel 2-bit PTQ framework that effectively addresses the unique challenges of quantizing dLLMs. By introducing Masked Calibration Simulation to handle timestep-dependent inputs, and developing a Data-aware Any-order Quantizer and an importance-guided Adaptive Blockwise Mixed Precision strategy, our method significantly mitigates quantization errors. Extensive experiments demonstrate that Quant-dLLM consistently outperforms SOTA 2-bit PTQ methods, establishing a new SOTA in 2-bit weight-only quantization for dLLMs and enabling efficient deployment on resource-constrained environments.

## ACKNOWLEDGMENTS

This work is supported by the National Natural Science Foundation of China (62501386, 625B1025) and also sponsored by CCF-Tencent Rhino-Bird Open Research Fund.

## ETHICS STATEMENT

The research conducted in the paper conforms, in every respect, with the ICLR Code of Ethics.

## REPRODUCIBILITY STATEMENT

We have provided implementation details in Section 4. We will also release all the code and models.

## LLM USAGE STATEMENT

Large Language Models (LLMs) were used solely for polishing writing. They did not contribute to the research content or scientific findings of this work.

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
