# QUANT-DLLM: POST-TRAINING EXTREME LOW-BIT QUANTIZATION FOR DIFFUSION LARGE LANGUAGE MODELS

CONTENTS

# A    FIRST-ORDER DAQ

The First-order Data-aware Any-order Quantizer (DAQ) aims to find a 1-bit approximation of a weight matrix by minimizing a quantization error that is weighted by the importance of each parameter. This method acknowledges that not all weights contribute equally to the model's performance and thus prioritizes the fidelity of more salient weights during quantization.

The objective function, consistent with the main text, is to minimize the weighted Frobenius norm of the quantization error:

$$\mathcal{L} = \left\| \mathbf{\Lambda} \odot (\mathbf{W} - \widehat{\mathbf{W}}) \right\|_F^2 \tag{1}$$

where $\mathbf{W} \in \mathbb{R}^{n \times m}$ is the full-precision weight matrix, $\mathbf{\Lambda} \in \mathbb{R}^{n \times m}$ is the importance mask derived from calibration data, and $\odot$ denotes the Hadamard (element-wise) product. The quantized weight matrix, $\widehat{\mathbf{W}}$, is parameterized as follows:

$$\widehat{\mathbf{W}} = (\boldsymbol{\alpha}^r (\boldsymbol{\alpha}^c)^\top) \odot \mathbf{B} \tag{2}$$

Here, $\boldsymbol{\alpha}^r \in \mathbb{R}^n$ and $\boldsymbol{\alpha}^c \in \mathbb{R}^m$ are the row and column scaling factor vectors, respectively, and $\mathbf{B} \in \{-1, +1\}^{n \times m}$ is the binary matrix.

The complete algorithm for the first-order DAQ is detailed in Algorithm 1. The process consists of three main stages: initialization, iterative refinement of scaling factors, and an optional re-rounding of the binary matrix.

**1. Initialization**    The algorithm begins by establishing an initial approximation of the full-precision weights $\mathbf{W}$. This involves initializing the binary matrix $\mathbf{B}$ and the scaling factors $\boldsymbol{\alpha}^r$ and $\boldsymbol{\alpha}^c$.

First, the binary matrix $\mathbf{B}$ is determined by the sign of the original weights, capturing their fundamental polarity:

$$\mathbf{B} = \mathrm{sign}(\mathbf{W}) \tag{3}$$

Next, inspired by ARB-LLM, the row and column scaling factors are initialized to approximate the magnitude of the weights. The row scaling factor $\boldsymbol{\alpha}^r$ is initialized as the mean absolute value of each row:

$$\boldsymbol{\alpha}^r = \frac{1}{m} |\mathbf{W}| \, \mathbf{1}_m \tag{4}$$

where $|\mathbf{W}|$ is the element-wise absolute value of $\mathbf{W}$ and $\mathbf{1}_m$ is an all-ones vector of size $m$. Subsequently, the column scaling factor $\boldsymbol{\alpha}^c$ is initialized based on the row-scaled weights:

$$\boldsymbol{\alpha}^c = \frac{1}{n} |\mathbf{W}|^\top (\boldsymbol{\alpha}_r)^{\circ-1} \tag{5}$$

where $(\boldsymbol{\alpha}_r)^{\circ-1}$ denotes the element-wise inverse of $\boldsymbol{\alpha}^r$. This initialization provides a reasonable starting point for the subsequent iterative optimization.

**2. RSR**    Following initialization, the algorithm enters an iterative loop to refine the scaling factors $\boldsymbol{\alpha}^r$ and $\boldsymbol{\alpha}^c$ for a total of $T$ rounds using the Row-column Successive Re-scaling (RSR) procedure. Within each iteration, we alternately update the scaling factors. Let $\mathbf{\Lambda}^2 = \mathbf{\Lambda} \odot \mathbf{\Lambda}$ be the element-wise square of the importance mask.

The $update\_\alpha_r$ function solves $\nabla_{\boldsymbol{\alpha}^r} \mathcal{L} = \mathbf{0}$ and updates the entire $\boldsymbol{\alpha}^r$ vector in closed form:

$$\boldsymbol{\alpha}^r \leftarrow \left[ (\mathbf{\Lambda}^2 \odot \mathbf{W} \odot \mathbf{B}) \boldsymbol{\alpha}^c \right] \oslash \left[ \mathbf{\Lambda}^2 (\boldsymbol{\alpha}^c \odot \boldsymbol{\alpha}^c) + \varepsilon \mathbf{1}_n \right] \tag{6}$$

where $\oslash$ denotes element-wise division and $\varepsilon$ is a small constant for numerical stability.

Similarly, the $update\_\alpha_c$ function solves $\nabla_{\boldsymbol{\alpha}^c} \mathcal{L} = \mathbf{0}$ using the newly updated $\boldsymbol{\alpha}^r$:

$$\boldsymbol{\alpha}^c \leftarrow \left[ (\mathbf{\Lambda}^2 \odot \mathbf{W} \odot \mathbf{B})^\top \boldsymbol{\alpha}^r \right] \oslash \left[ (\mathbf{\Lambda}^2)^\top (\boldsymbol{\alpha}^r \odot \boldsymbol{\alpha}^r) + \varepsilon \mathbf{1}_m \right] \tag{7}$$

This alternating update process monotonically decreases the data-aware objective function $\mathcal{L}$, progressively improving the approximation of the quantized matrix $\widehat{\mathbf{W}}$.

**3. Optional Re-rounding of B**    After updating the scaling factors in an iteration, the binary matrix $\mathbf{B}$ can be optionally re-evaluated. While holding $\mathbf{B}$ fixed during iterations simplifies optimization,

---

**Algorithm 1** DAQ[1]: First-order Data-aware Quantizer

---

func $\mathrm{DAQ}^1(\mathbf{W}, \mathbf{Z}, T, \lambda, \varepsilon)$

**Input** $\mathbf{W} \in \mathbb{R}^{n \times m}, \mathbf{Z} \in \mathbb{R}^{n \times m}, T \in \mathbb{N}, \lambda > 1,$
$\varepsilon > 0$

**Output** $\hat{\mathbf{W}} \in \mathbb{R}^{n \times m}, \boldsymbol{\alpha}_r \in \mathbb{R}^n, \boldsymbol{\alpha}_c \in \mathbb{R}^m,$
$\mathbf{B} \in \{\pm 1\}^{n \times m}$

1: $\boldsymbol{\Lambda} := \text{build\_importance\_mask}(\mathbf{Z}, \lambda)$
2: $\boldsymbol{\alpha}_r, \boldsymbol{\alpha}_c, \mathbf{B} := \text{binary\_rc\_init}(\mathbf{W})$
3: **for** $t = 1, 2, \ldots, T$ **do**
4: $\quad \boldsymbol{\alpha}_r \leftarrow \text{update\_}\alpha^{\mathrm{r}}(\mathbf{W}, \mathbf{B}, \boldsymbol{\alpha}_c, \boldsymbol{\Lambda}, \varepsilon)$
5: $\quad \boldsymbol{\alpha}_c \leftarrow \text{update\_}\alpha^{\mathrm{c}}(\mathbf{W}, \mathbf{B}, \boldsymbol{\alpha}_r, \boldsymbol{\Lambda}, \varepsilon)$
6: $\quad$ *optional* $\mathbf{B} \leftarrow \text{sign}(\mathbf{W})$
7: **end for**
8: $\hat{\mathbf{W}} := (\boldsymbol{\alpha}_r \boldsymbol{\alpha}_c^\top) \odot \mathbf{B}$
9: **return** $\hat{\mathbf{W}}, \boldsymbol{\alpha}_r, \boldsymbol{\alpha}_c, \mathbf{B}$

func binary\_rc\_init$(\mathbf{W})$

1: $\mathbf{B} := \text{sign}(\mathbf{W})$
2: $\boldsymbol{\alpha}_r := \frac{1}{m} |\mathbf{W}| \mathbf{1}_m$
3: $\boldsymbol{\alpha}_c := \frac{1}{n} |\mathbf{W}|^\top (\boldsymbol{\alpha}_r)^{\circ -1}$
4: **return** $\boldsymbol{\alpha}_r, \boldsymbol{\alpha}_c, \mathbf{B}$

**func** update\_$\alpha^{\mathrm{r}}(\mathbf{X}, \quad \mathbf{B}, \quad \boldsymbol{\alpha}_c, \quad \boldsymbol{\Lambda}, \quad \varepsilon)$

1: $\boldsymbol{\Lambda}^{(2)} := \boldsymbol{\Lambda} \odot \boldsymbol{\Lambda}$
2: $\mathbf{u} := (\boldsymbol{\Lambda}^{(2)} \odot \mathbf{X} \odot \mathbf{B}) \boldsymbol{\alpha}_c$
3: $\mathbf{v} := \boldsymbol{\Lambda}^{(2)} (\boldsymbol{\alpha}_c^{\circ 2}) + \varepsilon \mathbf{1}_n$
4: **return** $\mathbf{u} \oslash \mathbf{v}$

**func** update\_$\alpha^{\mathrm{c}}(\mathbf{X}, \quad \mathbf{B}, \quad \boldsymbol{\alpha}_r, \quad \boldsymbol{\Lambda}, \quad \varepsilon)$

1: $\boldsymbol{\Lambda}^{(2)} := \boldsymbol{\Lambda} \odot \boldsymbol{\Lambda}$
2: $\mathbf{u} := (\boldsymbol{\Lambda}^{(2)} \odot \mathbf{X} \odot \mathbf{B})^\top \boldsymbol{\alpha}_r$
3: $\mathbf{v} := (\boldsymbol{\Lambda}^{(2)})^\top (\boldsymbol{\alpha}_r^{\circ 2}) + \varepsilon \mathbf{1}_m$
4: **return** $\mathbf{u} \oslash \mathbf{v}$

**func** build\_importance\_mask$(\mathbf{Z}, \lambda)$

1: $\mu := \text{mean}(\mathbf{Z}), \quad \sigma := \text{std}(\mathbf{Z})$
2: $\mathbf{M} := \mathbb{I}(|\mathbf{Z} - \mu| > 3\sigma)$
3: **return** $\mathbf{1}_{n \times m} + (\lambda - 1) \mathbf{M}$

---

re-rounding it can sometimes further reduce quantization error. The optimal binary matrix that minimizes the objective is determined by the sign of the weights:

$$\mathbf{B} \leftarrow \text{sign}(\mathbf{W}) \tag{8}$$

In practice, for the first-order case, this optional step simply re-aligns the binary matrix with the original weights, reinforcing the initial structure. This is particularly useful if the iterative scaling process causes a significant shift in the reconstructed magnitudes.

# B  K-ORDER DAQ

To further enhance the representational capacity of the quantizer under a fixed bit-budget, the DAQ framework is extended to a K-order representation. This approach approximates the full-precision weight matrix $\mathbf{W}$ as a sum of $K$ distinct first-order components. Each component consists of its own binary matrix $\mathbf{B}^{(k)}$ and a corresponding pair of row-column scaling factors, $\boldsymbol{\alpha}_r^{(k)}$ and $\boldsymbol{\alpha}_c^{(k)}$. The K-order quantized weight matrix $\widehat{\mathbf{W}}$ is formulated as:

$$\hat{\mathbf{W}} = \sum_{k=1}^{K} \left( \boldsymbol{\alpha}_r^{(k)} (\boldsymbol{\alpha}_c^{(k)})^\top \right) \odot \mathbf{B}^{(k)} \tag{9}$$

The optimization for this K-order representation is performed through a greedy, iterative process, as detailed in Algorithm 2. The procedure can be broken down into three primary stages: greedy initialization, iterative refinement of all components, and a final update of the binary matrices.

## B.1  ALGORITHMIC PROCESS OF K-ORDER DAQ

**1. Greedy Initialization** The algorithm begins with a greedy, stage-wise construction of the $K$ components. This process iteratively fits a new first-order component to the residual error from the previous stages. For $k = 1, 2, \ldots, K$:

1. **Compute Current Residual Matrix:** The residual matrix $\mathbf{R}^{(k)}$ is the difference between the original weights and the sum of all previously initialized components.

$$\mathbf{R}^{(k)} = \mathbf{W} - \sum_{q=1}^{k-1} \left( \left( \boldsymbol{\alpha}_r^{(q)} (\boldsymbol{\alpha}_c^{(q)})^\top \right) \odot \mathbf{B}^{(q)} \right) \tag{10}$$

For the first iteration ($k = 1$), the residual is the full-precision weight matrix itself, $\mathbf{R}^{(1)} = \mathbf{W}$.

2. **Initialize the $k$-th Component:** The $binary\_rc\_init$ function is called on the current residual to obtain the initial parameters for the $k$-th component.

$$\boldsymbol{\alpha}_r^{(k)}, \boldsymbol{\alpha}_c^{(k)}, \mathbf{B}^{(k)} := \mathrm{binary\_rc\_init}(\mathbf{R}^{(k)}) \tag{11}$$

The internal operations of this function are:

$$\mathbf{B}^{(k)} = \mathrm{sign}(\mathbf{R}^{(k)}) \tag{12}$$

$$\boldsymbol{\alpha}_r^{(k)} = \tfrac{1}{m} \, |\mathbf{R}^{(k)}| \, \mathbf{1}_m \tag{13}$$

$$\boldsymbol{\alpha}_c^{(k)} = \tfrac{1}{n} \, |\mathbf{R}^{(k)}|^\top \, (\boldsymbol{\alpha}_r^{(k)})^{\circ -1} \tag{14}$$

This greedy procedure provides a strong starting point for the joint optimization phase.

**2. Joint RSR** After initializing all $K$ components, the algorithm enters a main iterative loop to jointly refine all parameters for $T$ rounds. In each round, the algorithm cycles through every component $k = 1, \ldots, K$ and refines its scaling factors while keeping all other components fixed. For each component $k$:

1. **Compute Partial Residual Matrix:** The partial residual $\mathbf{R}^{(k)}$ is calculated by subtracting all components *except* the $k$-th one from the original weights. This isolates the target for the current component's optimization.

$$\mathbf{R}^{(k)} = \mathbf{W} - \sum_{q \neq k} \left( \left( \boldsymbol{\alpha}_r^{(q)} (\boldsymbol{\alpha}_c^{(q)})^\top \right) \odot \mathbf{B}^{(q)} \right) \tag{15}$$

2. **Update Row and Column Scaling Factors:** The RSR procedure is applied to find the optimal scaling factors for the $k$-th component that best approximate the partial residual $\mathbf{R}^{(k)}$. Let $\boldsymbol{\Lambda}^2 = \boldsymbol{\Lambda} \odot \boldsymbol{\Lambda}$.

$$\boldsymbol{\alpha}_r^{(k)} \leftarrow \left[ (\boldsymbol{\Lambda}^2 \odot \mathbf{R}^{(k)} \odot \mathbf{B}^{(k)}) \boldsymbol{\alpha}_c^{(k)} \right] \oslash \left[ \boldsymbol{\Lambda}^2 (\boldsymbol{\alpha}_c^{(k)} \odot \boldsymbol{\alpha}_c^{(k)}) + \varepsilon \mathbf{1}_n \right] \tag{16}$$

$$\boldsymbol{\alpha}_c^{(k)} \leftarrow \left[ (\boldsymbol{\Lambda}^2 \odot \mathbf{R}^{(k)} \odot \mathbf{B}^{(k)})^\top \boldsymbol{\alpha}_r^{(k)} \right] \oslash \left[ (\boldsymbol{\Lambda}^2)^\top (\boldsymbol{\alpha}_r^{(k)} \odot \boldsymbol{\alpha}_r^{(k)}) + \varepsilon \mathbf{1}_m \right] \tag{17}$$

This process allows the parameters of all components to be jointly optimized to minimize the overall quantization error $\mathcal{L}$.

**3. Update of Binary Matrices** After the iterative refinement of the scaling factors within a round, the binary matrices $\{\mathbf{B}^{(k)}\}_{k=1}^K$ are jointly updated. Since the binary values are discrete, this is solved via an exhaustive search for each weight position $(i, j)$.

1. **Construct Scale Vector:** For each position $(i, j)$, a vector $\mathbf{s}_{ij} \in \mathbb{R}^K$ is constructed, containing the scaling products from all $K$ components.

$$\mathbf{s}_{ij} = \begin{bmatrix} (\alpha_{r,i}^{(1)})(\alpha_{c,j}^{(1)}) \\ \vdots \\ (\alpha_{r,i}^{(K)})(\alpha_{c,j}^{(K)}) \end{bmatrix} \tag{18}$$

2. **Minimal-Error Search:** The $update\_B$ function finds the optimal binary vector $\mathbf{b}_{ij} \in \{-1, +1\}^K$ that minimizes the reconstruction error for $W_{ij}$.

$$\mathbf{b}_{ij} = \arg \min_{\mathbf{b} \in \{-1, +1\}^K} \left| W_{ij} - \mathbf{s}_{ij}^\top \mathbf{b} \right| \tag{19}$$

The elements of the resulting vector $\mathbf{b}_{ij} = [b_1, \ldots, b_K]^\top$ are then used to update the corresponding entries in the binary matrices:

$$(\mathbf{B}^{(k)})_{ij} \leftarrow b_k, \quad \text{for } k = 1, \ldots, K \tag{20}$$

This step ensures that the binary carriers are optimally aligned with the refined scaling factors after each round of optimization.

---

**Algorithm 2** DAQ: Data-aware any-order Quantizer

---

func DAQ$(\mathbf{W}, \mathbf{Z}, K, T, \lambda)$
**Input** $\mathbf{W}, \mathbf{Z} \in \mathbb{R}^{n \times m}, K, T \in \mathbb{N}, \lambda > 1$
**Output**

$$\hat{\mathbf{W}} = \sum_{k=1}^{K} \left( \boldsymbol{\alpha}_r^{(k)} \, \boldsymbol{\alpha}_c^{(k)\top} \right) \odot \mathbf{B}^{(k)}$$

1: $\boldsymbol{\Lambda} := \text{build\_importance\_mask}(\mathbf{Z}, \lambda)$
2: $\hat{\mathbf{W}} := \mathbf{0}_{n \times m}$
3: **for** $k = 1, 2, \ldots, K$ **do**
4:    $\mathbf{R} \leftarrow \mathbf{W} - \hat{\mathbf{W}}$
5:    $\boldsymbol{\alpha}_r^{(k)}, \boldsymbol{\alpha}_c^{(k)}, \mathbf{B}^{(k)} \leftarrow \text{binary\_rc\_init}(\mathbf{R})$
6:    $\mathbf{S}^{(k)} \leftarrow \boldsymbol{\alpha}_r^{(k)} \big( \boldsymbol{\alpha}_c^{(k)} \big)^\top$
7:    $\hat{\mathbf{W}} \leftarrow \hat{\mathbf{W}} + \mathbf{S}^{(k)} \odot \mathbf{B}^{(k)}$
8: **end for**
9: **for** $t = 1, 2, \ldots, T$ **do**
10:    **for** $k = 1, 2, \ldots, K$ **do**
11:       $\hat{\mathbf{W}}_{\setminus k} \leftarrow \sum_{q \neq k} \big( \mathbf{S}^{(q)} \odot \mathbf{B}^{(q)} \big)$
12:       $\mathbf{R}^{(k)} \leftarrow \mathbf{W} - \hat{\mathbf{W}}_{\setminus k}$
13:       $\boldsymbol{\alpha}_r^{(k)} \leftarrow \text{update\_}\alpha_\text{r}(\mathbf{R}^{(k)}, \mathbf{B}^{(k)}, \boldsymbol{\alpha}_c^{(k)}, \boldsymbol{\Lambda})$
14:       $\boldsymbol{\alpha}_c^{(k)} \leftarrow \text{update\_}\alpha_\text{c}(\mathbf{R}^{(k)}, \mathbf{B}^{(k)}, \boldsymbol{\alpha}_r^{(k)}, \boldsymbol{\Lambda})$
15:       $\mathbf{S}^{(k)} \leftarrow \boldsymbol{\alpha}_r^{(k)} \big( \boldsymbol{\alpha}_c^{(k)} \big)^\top$
16:    **end for**
     $\{\mathbf{B}^{(k)}\}_{k=1}^{K} \leftarrow$
17:    $\text{update\_B}\Big( \mathbf{W}, \{\boldsymbol{\alpha}_r^{(k)}\}_{k=1}^{K}, \{\boldsymbol{\alpha}_c^{(k)}\}_{k=1}^{K} \Big)$
18:    $\hat{\mathbf{W}} \leftarrow \sum_{k=1}^{K} \mathbf{S}^{(k)} \odot \mathbf{B}^{(k)}$
19: **end for**
20: **return** $\hat{\mathbf{W}}$

func binary\_rc\_init$(\mathbf{X})$

1: $\boldsymbol{\alpha}_r \leftarrow \frac{1}{m} |\mathbf{X}| \, \mathbf{1}_m$
2: $\boldsymbol{\alpha}_c \leftarrow \frac{1}{n} |\mathbf{X}|^\top \text{diag}(\boldsymbol{\alpha}_r)^{-1} \, \mathbf{1}_n$
3: $\mathbf{B} \leftarrow \text{sign}(\mathbf{X})$
4: **return** $\boldsymbol{\alpha}_r, \, \boldsymbol{\alpha}_c, \, \mathbf{B}$

func update\_$\alpha_\text{r}(\mathbf{X}, \mathbf{B}, \boldsymbol{\alpha}_c, \boldsymbol{\Lambda})$

1: $\mathbf{u} \leftarrow (\boldsymbol{\Lambda}^2 \odot \mathbf{X} \odot \mathbf{B}) \, \boldsymbol{\alpha}_c$
2: $\mathbf{v} \leftarrow (\boldsymbol{\Lambda}^2 (\text{diag}\,\boldsymbol{\alpha}_c) \, \boldsymbol{\alpha}_c) + \varepsilon \, \mathbf{1}_n$
3: **return** $\mathbf{u} \oslash \mathbf{v}$

func update\_$\alpha_\text{c}(\mathbf{X}, \mathbf{B}, \boldsymbol{\alpha}_r, \boldsymbol{\Lambda})$

1: $\mathbf{u} \leftarrow (\boldsymbol{\Lambda}^2 \odot \mathbf{X} \odot \mathbf{B})^\top \boldsymbol{\alpha}_r$
2: $\mathbf{v} \leftarrow (\boldsymbol{\Lambda}^2)^\top (\text{diag}(\boldsymbol{\alpha}_r) \, \boldsymbol{\alpha}_r) + \varepsilon \, \mathbf{1}_m$
3: **return** $\mathbf{u} \oslash \mathbf{v}$

func update\_B$\Big( \mathbf{W}, \{\boldsymbol{\alpha}_r^{(k)}\}_{k=1}^{K}, \{\boldsymbol{\alpha}_c^{(k)}\}_{k=1}^{K} \Big)$

1: **for** $i = 1, \ldots, n$ **do**
2:    **for** $j = 1, \ldots, m$ **do**
3:       $\mathbf{s} \leftarrow \big[ (\alpha_r^{(k)})_i (\alpha_c^{(k)})_j \big]_{k=1}^{K}$
4:       $\mathbf{b} \leftarrow \text{search}\big( W_{ij}, \mathbf{s} \big)$
5:       **for** $k = 1, \ldots, K$ **do**
6:          $(\mathbf{B}^{(k)})_{ij} \leftarrow \mathbf{b}_k$
7:       **end for**
8:    **end for**
9: **end for**
10: **return** $\{\mathbf{B}^{(k)}\}_{k=1}^{K}$

func build\_importance\_mask$(\mathbf{Z}, \lambda)$

1: $\mu \leftarrow \text{mean}(\mathbf{Z}), \quad \sigma \leftarrow \text{std}(\mathbf{Z})$
2: $\mathbf{M} \leftarrow \mathbb{I}\big( |\mathbf{Z} - \mu| > 3\sigma \big)$
3: **return** $\mathbf{1}_{n \times m} + (\lambda - 1) \, \mathbf{M}$

---

## C    JUSTIFICATION FOR THE DATA-AWARE FROBENIUS PROXY

In our Quant-dLLM framework, the ideal objective is to directly minimize the quantization error on the layer's output. This is captured by the data-aware objective $\mathcal{L}_2$, which utilizes the second-moment matrix $\mathbf{S}_{\mathrm{MCS}}$ computed from the calibration data generated by our Masked Calibration Simulation (MCS) method:

$$\mathcal{L}_2 = \mathrm{Tr}\left((\mathbf{W} - \widehat{\mathbf{W}})\,\mathbf{S}_{\mathrm{MCS}}\,(\mathbf{W} - \widehat{\mathbf{W}})^\top\right) \tag{21}$$

While this objective is theoretically superior, its direct optimization with our separable row-column (RC) scaling, $\widehat{\mathbf{W}} = (\boldsymbol{\alpha}^r(\boldsymbol{\alpha}^c)^\top) \odot \mathbf{B}$, is computationally intractable. This section justifies our use of a tractable proxy.

### C.1    THE CHALLENGE: PARAMETER COUPLING IN RSR

The primary difficulty arises from **parameter coupling** when attempting to derive an update rule for the column scaling factors $\boldsymbol{\alpha}^c$ within our Row-column Successive Re-scaling (RSR) algorithm. When we take the partial derivative of $\mathcal{L}_2$ with respect to a single column scaling factor, $\alpha_{c,t}$, and set it to zero, the resulting equation for $\alpha_{c,t}$ remains dependent on all other column scaling factors $\alpha_{c,k}$ (where $k \neq t$).

This coupling is illustrated by the conceptual structure of the resulting equation:

$$\frac{\partial \mathcal{L}_2}{\partial \alpha_{c,t}} \propto \sum_{i=1}^{n} \alpha_{r,i}^2 B_{it}^2 \sum_{k=1}^{m} S_{tk}\alpha_{c,k} - \sum_{i=1}^{n} \alpha_{r,i} B_{it} \sum_{k=1}^{m} W_{ik}S_{kt} = 0 \tag{22}$$

The term $\sum_{k=1}^{m} S_{tk}\alpha_{c,k}$ clearly demonstrates the problem: due to the non-diagonal nature of the $\mathbf{S}_{\mathrm{MCS}}$ matrix (where $S_{tk}$ can be non-zero for $t \neq k$), the optimal value for $\alpha_{c,t}$ is part of a system of linear equations involving the entire vector $\boldsymbol{\alpha}^c$. Solving this dense system at each step is computationally prohibitive for a PTQ method.

### C.2    THE SOLUTION: DATA-AWARE OBJECTIVE REFORMULATION (DOR)

To overcome this challenge, we introduced the **Data-aware Objective Reformulation (DOR)** in the main paper. DOR serves as a bridge, creating a computationally efficient yet effective proxy for the intractable $\mathcal{L}_2$ objective. By analyzing the importance matrix $\mathbf{Z}$ derived from $\mathbf{S}_{\mathrm{MCS}}$, DOR constructs a sparse importance mask $\boldsymbol{\Lambda}$.

This allows us to optimize the **data-aware Frobenius proxy**:

$$\widehat{\mathcal{L}}_{\boldsymbol{\Lambda}}(\boldsymbol{\alpha}_r, \boldsymbol{\alpha}_c) = \left\|\boldsymbol{\Lambda} \odot (\mathbf{W} - \hat{\mathbf{W}})\right\|_F^2 \tag{23}$$

This objective successfully retains the critical data-dependent information (by up-weighting salient elements) while being structurally suitable for our efficient, closed-form RSR updates.

## D    MASKED-DIFFUSION DECODING

**Gaussian masked diffusion (MDM).**    For a continuous input $x$ and a binary mask $r \in \{0,1\}^L$, we perturb only the masked coordinates to obtain $x_t$ and train a noise predictor $\varepsilon_\theta(x_t, r, t)$ with the standard denoising objective

$$\mathcal{L}_{\mathrm{MDM}} = \mathbb{E}_{x,\,r,\,\varepsilon,\,t} \left\|\varepsilon - \varepsilon_\theta(x_t, r, t)\right\|_2^2, \quad \varepsilon \sim \mathcal{N}(0, I). \tag{24}$$

**Absorbing-state discrete variant.**    For token sequences $y \in \{e_1, \ldots, e_K\}^L$ with absorbing token $M$ (one-hot $m$) and a monotone visibility schedule $\alpha_t \in [0,1]$ (we use a linear schedule by default), the forward corruption is

$$q(y_t \mid y) = \mathrm{Cat}\big(\alpha_t\, y + (1 - \alpha_t)\, m\big). \tag{25}$$

Decoding proceeds from $t=1$ to $t=0$ in $T$ steps. Let $\boldsymbol{\pi}_\theta(y_t) = \mathrm{softmax}\big(f_\theta(y_t)\big)$ and $0 \leq s < t \leq 1$ with $s, t \in \{0, \frac{1}{T}, \ldots, 1\}$. Define

$$\lambda_{s|t} = \frac{\alpha_s - \alpha_t}{1 - \alpha_t} \in [0,1]. \tag{26}$$

Then each position $i$ evolves by the copy/denoise rule

$$\textbf{copy:} \quad y_t^{(i)} \neq M \Rightarrow y_s^{(i)} = y_t^{(i)}, \qquad \textbf{denoise:} \quad y_t^{(i)} = M \Rightarrow y_s^{(i)} \sim (1 - \lambda_{s|t}) \, m + \lambda_{s|t} \, \boldsymbol{\pi}_\theta(y_t). \tag{27}$$

In practice we perform parallel position updates. Optional heuristics (e.g., blockwise semi-autoregressive ordering, confidence-based remasking) can be added without changing Eqs. equation 25–equation 27.

## E DETAILED EXPERIMENTS SETTINGS

Our evaluation is conducted using a framework adapted from the `lm-evaluation-harness`, tailored for diffusion language models. As mentioned in the main text, our benchmarks are grouped into three categories: general-knowledge QA, mathematical and scientific reasoning, and code generation. The specific settings for each model family are detailed below. For all code generation tasks, the `HF_ALLOW_CODE_EVAL=1` environment variable is set to enable execution-based evaluation.

### E.1 SETTINGS FOR LLaDA MODELS

For the LLaDA model series (LLaDA-8B-Base, LLaDA-8B-Instruct, and LLaDA-1.5), we apply a consistent set of hyperparameters for each evaluation task to ensure a fair comparison.

**General-knowledge QA.** For tasks based on conditional likelihood estimation, we use 128 Monte Carlo samples ($mc_{num} = 128$) unless specified otherwise. The task-specific settings are as follows:

- **MMLU:** We use a 5-shot setting with a Classifier-Free Guidance (CFG) scale of 0.0. For our ablation runs, $mc_{num}$ was set to 1 for faster evaluation.
- **WinoGrande:** We use a 5-shot setting with a CFG scale of 0.0.
- **HellaSwag, ARC-Easy, ARC-Challenge:** These are evaluated in a 0-shot setting with a CFG scale of 0.5.
- **PIQA:** This is evaluated in a 0-shot setting with a CFG scale of 0.5.

**Mathematical & Scientific Reasoning and Code Generation.** For tasks requiring conditional generation, the settings are configured to balance performance and generation quality.

- **GSM8K:** We employ diffusion semi-autoregressive sampling by setting $gen_{length} = 256$ and a $block_{length} = 32$, with 256 denoising steps. This approach is effective for complex, multi-step reasoning.
- **BBH:** This task is evaluated with standard parallel decoding, where $gen_{length}$, $steps$, and $block_{length}$ are all set to 128.
- **GPQA:** This task is evaluated as a likelihood estimation problem in a 0-shot setting, with a batch size of 1, $mc_{num} = 128$, and a CFG scale of 0.5.
- **HumanEval & MBPP:** Both code generation tasks are evaluated with standard parallel decoding, where $gen_{length}$, $steps$, and $block_{length}$ are all set to 128.

### E.2 SETTINGS FOR DREAM MODELS

For the Dream model series (Dream-7B-Base and Dream-7B-Instruct), a similar evaluation protocol is followed to maintain consistency. The primary difference is in the number of few-shot examples for some tasks, as detailed in the evaluation script.

**General-knowledge QA.** Like the LLaDA series, these tasks are evaluated using 128 Monte Carlo samples. The few-shot settings are as follows:

- **MMLU, WinoGrande:** Evaluated in a 5-shot setting.
- **ARC-Easy, ARC-Challenge, HellaSwag, PIQA:** All evaluated in a 0-shot setting.

Table 1: Effectiveness of MCS on previous methods. We report MMLU in 5 shots.

((a)) LLaDA-8B-Base

| Model | MCS | Method | MMLU(5) ↑ |
|---|---|---|---|
| | ✗ | GPTQ | 34.75 |
| | ✓ | GPTQ | 36.75 |
| LLaDA-8B-Base | ✗ | GPTAQ | 34.73 |
| | ✓ | GPTAQ | 37.92 |
| | ✗ | Slim-LLM | 47.98 |
| | ✓ | Slim-LLM | 50.11 |

((b)) Dream-7B-Base

| Model | MCS | Method | MMLU(5) ↑ |
|---|---|---|---|
| | ✗ | GPTQ | 23.84 |
| | ✓ | GPTQ | 24.08 |
| Dream-7B-Base | ✗ | GPTAQ | 23.90 |
| | ✓ | GPTAQ | 24.68 |
| | ✗ | Slim-LLM | 25.54 |
| | ✓ | Slim-LLM | 26.01 |

**Mathematical & Scientific Reasoning and Code Generation.** For generative tasks, the Dream models are evaluated using the same general protocol as LLaDA.

# F MORE EXPERIMENTS

## F.1 EFFECTIVENESS OF MCS ON BASELINE METHODS

To demonstrate the general applicability of our Masked Calibration Simulation (MCS), we investigate its effectiveness when applied to existing state-of-the-art (SOTA) post-training quantization methods. We integrate MCS into the calibration stage of GPTQ, GPTAQ, and Slim-LLM, and evaluate the impact on MMLU 5-shot accuracy for the LLaDA-8B-Base and Dream-7B-Base models. The results are presented in Table 1.

As shown in Table 1(a), applying MCS consistently improves the performance of all baseline methods on LLaDA-8B-Base. Specifically, the MMLU score for GPTQ increases from 34.75% to 36.75%, and GPTAQ sees a significant boost from 34.73% to 37.92%. Similarly, the performance of the strong baseline, Slim-LLM, is enhanced from 47.98% to 50.11%.

A similar trend is observed on the Dream-7B-Base model, as detailed in Table 1(b). While the improvements are more modest on this model, MCS still provides consistent gains across the board. For instance, GPTAQ's accuracy rises from 23.90% to 24.68%, and Slim-LLM's score improves from 25.54% to 26.01%.

These results strongly indicate that MCS is not limited to our DAQ framework but serves as a valuable, standalone module for quantizing dLLMs. By aligning the calibration data distribution with the model's native diffusion-denoising process, MCS effectively reduces the calibration-inference mismatch and boosts the performance of various PTQ techniques.