# OpenReview forum: "Quant-dLLM: Post-Training Extreme Low-Bit Quantization for Diffusion Large Language Models"
_ICLR.cc/2026/Conference — ICLR 2026 Poster_

### Official Review · Reviewer_hDfx · 2025-10-29

**Soundness:** 3
**Presentation:** 3
**Contribution:** 3
**Rating:** 6
**Confidence:** 2

**Summary:**

This paper introduces Quant-dLLM, a post-training quantization (PTQ) framework tailored for diffusion-based large language models (dLLMs) under an extreme 2-bit weight-only budget. The authors propose three key components: 1) Masked Calibration Simulation (MCS) to align calibration with timestep-dependent masking; 2)Data-aware Any-order Quantizer (DAQ) to reconstruct weights using multi-binary matrices with row-column scaling; 3)Adaptive Blockwise Mixed Precision (ABMP) to allocate varying precision across blocks under a strict 2-bit average.
Extensive experiments on recent dLLMs (LLaDA, Dream) show that Quant-dLLM significantly outperforms strong baselines (GPTQ, GPTAQ, Slim-LLM) across general QA, math, and code tasks.

**Strengths:**

1. The paper is well-written, with detailed algorithms and clear exposition. The authors commit to releasing code and models.
2. Quant-dLLM shows improvements of challenging tasks like math and code. Ablation studies verify the effectiveness of each module.
3. MCS directly tackles the distributional mismatch caused by masked denoising in dLLMs, a unique challenge not present in autoregressive LLMs.

**Weaknesses:**

1. Missing Baselines: The paper focuses on 2-bit quantization and lacks classical comparison methods [1] [2].
2. Incremental Novelty: The core ideas, multi-binary approximation and mixed-precision allocation, are extensions of prior work (e.g., Slim-LLM). The authors should clearly articulate the technical distinctions and why these ideas are nontrivial to adapt to dLLMs.

[1] QuIP: 2-Bit Quantization of Large Language Models With Guarantees

[2] DB-LLM: Accurate Dual-Binarization for Efficient LLMs

**Questions:**

See Weaknesses.

---

> ### Author Response · Authors · 2025-11-23
> **Response to Reviewer hDfx (denoted as R5)**
>
> `Q5-1:`Missing Baselines: The paper focuses on 2-bit quantization and lacks classical comparison methods [1] [2].
> [1] QuIP: 2-Bit Quantization of Large Language Models With Guarantees
> [2] DB-LLM: Accurate Dual-Binarization for Efficient LLMs
>
> `A5-1:`Thank you for the suggestion. We now include QuIP, a representative 2-bit PTQ baseline, for a more complete comparison. As for DB-LLM, it is not a PTQ method, as it requires full fine-tuning with a distill training strategy rather than post-training quantization. This training-dependent pipeline makes it incompatible with the PTQ setting of our work, and therefore not directly comparable under the same constraints. As shown below, QuIP exhibits substantial degradation on LLaDA-8B-Base, while our Quant-dLLM achieves a higher accuracy under the same 2-bit constraint.
>
> | Method | ARC-C ↑ |
> |--------|---------|
> | FP16   | 0.4411  |
> | QuIP   | 0.2436  |
> | Ours   | 0.3626  |
>
> `Q5-2:`Incremental Novelty: The core ideas, multi-binary approximation and mixed-precision allocation, are extensions of prior work (e.g., Slim-LLM). The authors should clearly articulate the technical distinctions and why these ideas are nontrivial to adapt to dLLMs.
>
> `A5-2:`Thank you for the comment. Quant-dLLM is not a direct extension of Slim-LLM. First, we introduce Masked Calibration Simulation (MCS), which aligns quantization with the timestep-dependent masking behavior unique to diffusion LLMs; this adaptation is essential, since Slim-LLM’s AR-style calibration performs weakly under masked-denoising distributions. Second, we design a Data-aware Any-order Quantizer (DAQ) with weighted optimization that fits masked activations more accurately and yields much stronger 2-bit performance. Finally, while our bit-allocation strategy is inspired by Slim-LLM, our implementation is simpler and more efficient, making it more suitable for large-scale dLLMs.

---

> > ### Comment · Reviewer_hDfx · 2025-11-27
> > **Thank you for the response.**
> >
> > The response has alleviated my concerns. I will maintain my positive score with a confidence of 2.

---

> > > ### Author Response · Authors · 2025-12-02
> > >
> > > We sincerely thank Reviewer hDfx for the careful assessment. We are pleased that our responses helped clear up the concerns, and we appreciate your decision to retain the positive score in the submission.

---

### Official Review · Reviewer_ayMd · 2025-10-31

**Soundness:** 4
**Presentation:** 2
**Contribution:** 3
**Rating:** 4
**Confidence:** 3

**Summary:**

This paper proposes an ultra low bit PTQ method Quant-dLLM, which aims at addressing the specific characteristics in DLLM quantization including calibration, sensitivity distribution and low-bit quantization optimization. Extensive comparison and ablation results demonstrate its effectiveness.

**Strengths:**

By considering the unique characteristics of DLLMs, the authors identify the shortcomings of existing LLM quantization methods in this context and conduct targeted explorations to overcome them. As one of the pioneering studies in this direction, this work offers meaningful insights and guidance for subsequent research.

**Weaknesses:**

1. The paper requires improvements in writing, as it lacks a logical flow and several parts are inadequately explained, causing confusion for the readers.
2. The paper introduces block-wise bitwidth allocation but lacks a clear definition of what constitutes a *block*. As far as I understand, it should refer to the same concept as *group* in group-wise quantization, rather than a transformer block. This should be made consistent throughout the paper.
3. The importance matrix is unstructured and is related to the scaling factors and bitwidth allocation. If this information is required during inference, such as identifying the corresponding weights for the scaling factor when GEMM, the additional memory overhead introduced by the importance matrix will be non-negligible. For instance, an importance matrix could increase the average bitwidth of the weights by 1bit. The authors should provide a more detailed explanation of this aspect.

**Questions:**

1. The paper focuses on the 2-bit quantization but lacks specific 2-bit quantization baselines, such as QuIP[1] and PBLLM[2], which would provide a more comprehensive comparison.
2. In addition to accuracy, the authors should also provide memory usage and runtime statistics for the quantization process. Efficiency is an important factor that determines the broader impact of the proposed method

---

> [1] Chee J, Cai Y, Kuleshov V, et al. Quip: 2-bit quantization of large language models with guarantees[J]. Advances in Neural Information Processing Systems, 2023, 36: 4396-4429.

>[2] Shang Y, Yuan Z, Wu Q, et al. Pb-llm: Partially binarized large language models[J]. arXiv preprint arXiv:2310.00034, 2023.

---

> ### Author Response · Authors · 2025-11-23
> **Response to Reviewer ayMd (denoted as R4)**
>
> `Q4-1:`The paper requires improvements in writing, as it lacks a logical flow and several parts are inadequately explained, causing confusion for the readers.
>
> `A4-1:`We appreciate the reviewer’s feedback and apologize for the lack of clarity in the current draft. We will revise the writing to ensure the presentation is more coherent and easier to follow.
>
> `Q4-2:`The paper introduces block-wise bitwidth allocation but lacks a clear definition of what constitutes a block. As far as I understand, it should refer to the same concept as group in group-wise quantization, rather than a transformer block. This should be made consistent throughout the paper.
>
> `A4-2:`We apologize for the unclear terminology. In our method, a block refers to a group of n columns within a weight matrix, not a Transformer block. Throughout the paper, we use n = 128, which follows the common usage in block-wise quantization methods. This definition is consistent with prior work such as GPTQ [ref1].
> [ref1] https://arxiv.org/pdf/2210.17323
>
> `Q4-3:`The importance matrix is unstructured and is related to the scaling factors and bitwidth allocation. If this information is required during inference, such as identifying the corresponding weights for the scaling factor when GEMM, the additional memory overhead introduced by the importance matrix will be non-negligible. For instance, an importance matrix could increase the average bitwidth of the weights by 1bit. The authors should provide a more detailed explanation of this aspect.
>
> `A4-3:`Thank you for the question. The importance matrix is not stored or used during inference. It is only computed once during quantization to (i) assign block bit-widths and (ii) guide the optimization of \\(\\alpha_r\\) and \\(\\alpha_c\\). After quantization, it is discarded.
>
> The model only stores:
> - B (binary matrix),
> - \\(\\alpha_r\\) and \\(\\alpha_c\\) (FP16 vectors),
> - a small 2-bit list of block orders.
>
> No additional large matrices are kept, and the importance matrix does not introduce extra runtime memory overhead.
>
> `Q4-4:`The paper focuses on the 2-bit quantization but lacks specific 2-bit quantization baselines, such as QuIP[1] and PBLLM[2], which would provide a more comprehensive comparison.
> [1] Chee J, Cai Y, Kuleshov V, et al. Quip: 2-bit quantization of large language models with guarantees[J]. Advances in Neural Information Processing Systems, 2023, 36: 4396-4429.
> [2] Shang Y, Yuan Z, Wu Q, et al. Pb-llm: Partially binarized large language models[J]. arXiv preprint arXiv:2310.00034, 2023.
>
> `A4-4:`Thank you for the suggestion. We agree that including 2-bit baselines provides a more complete comparison. We therefore add results for QuIP and PB-LLM, which are two representative 2-bit quantization approaches. As shown below, both methods exhibit substantial degradation on LLaDA-8B-Base, while our Quant-dLLM achieves a higher accuracy under the same 2-bit constraint.
>
> | Method | ARC-C ↑ |
> |--------|---------|
> | FP16   | 0.4411  |
> | QuIP   | 0.2436  |
> | PB-LLM | 0.2534  |
> | Ours   | 0.3626  |
>
> `Q4-5:`In addition to accuracy, the authors should also provide memory usage and runtime statistics for the quantization process. Efficiency is an important factor that determines the broader impact of the proposed method.
>
> `A4-5:`Thank you for highlighting the importance of efficiency. We provide both memory usage and runtime statistics for the quantization process as follows. Using 128 calibration samples, quantizing LLaDA-8B-Base on a single NVIDIA A6000 (48 GB) requires about 37 GB of GPU memory and finishes in 32 minutes. We will include these measurements in the revised version to better reflect the practical cost of the method.

---

> ### Comment · Reviewer_ayMd · 2025-11-27
> **Response to authors**
>
> Thank you for your rebuttal. Can you provide the efficiency comparison as well as the other revision into the Appendix of revised PDF?

---

> > ### Author Response · Authors · 2025-12-02
> >
> > We thank Reviewer ayMd for the helpful follow-up comment. We appreciate your time, and we will add the efficiency comparison and other requested revisions to the supplementary materials in the last version.

---

### Official Review · Reviewer_xcW1 · 2025-11-01

**Soundness:** 4
**Presentation:** 4
**Contribution:** 3
**Rating:** 8
**Confidence:** 3

**Summary:**

The paper proposes Quant-dLLM, a 2-bit, weight-only PTQ pipeline designed explicitly for diffusion LLMs (dLLMs). It has three parts: first,  a masked calibration simulation (MCS)  that creates a synthetic dataset for the calibration of the quantizer appropriate for dLLM, seconds, a data-aware any-order quantizer (DAQ) inspired by DB-LLM that represents weights as a sum of row–column–scaled binary matrices, and finally, a mixed-precision approach inspired by Slim-LLM that allocates orders $K \in \{1,2,3\}$ per block under a strict 2-bit average budget. Across LLaDA and Dream series models, Quant-dLLM outperforms strong 2-bit weight-only PTQ baselines (GPTQ, GPTAQ, Slim-LLM) on general, math/science, and code tasks.

**Strengths:**

This is a well-motivated PTQ paper that clearly identifies the idiosyncrasies of diffusion LLMs (dLLMs) and proposes a data-aware combination of PTQ and mixed-precision quantization to aggressively compress models under a strict 2-bit average precision budget. The approach is conceptually elegant and technically well-executed, addressing a real gap between autoregressive and diffusion-style inference.

The results show consistent improvements across all benchmarks compared to existing low-bit PTQ baselines (GPTQ, GPTAQ, Slim-LLM), including on challenging math and code tasks. The paper is carefully presented, with clear motivation, ablation studies, and sound empirical analysis. Overall, this is a strong, mature piece of work that meaningfully advances post-training quantization for a new class of models.

**Weaknesses:**

* Limited analysis of runtime/latency and kernel costs. Quant-dLLM reports a favourable model size (Table 3) but lacks end-to-end latency/throughput on full models and a cost breakdown for DAQ/ABMP kernels. Reporting wall-clock metrics on common hardware and the proportion of time in quant ops would help assess deployability

* Sensitivity to hyperparameters is only partially explored:  λ (importance mask weight), 3σ thresholding, group/block sizes, or order K choices beyond the \{1,2,3\} scheme.

**Questions:**

* Section 4.4: any discussion or latency/throughput? Are there any costs associated with the quantisation method that would affect deployment, e.g. additional kernels, etc?`

* Ablation studies: I would like to see some ablation studies (even in the appendix) on the effect of the quantization block size or the effect of $\lambda$

* Line 320: regarding the bit relocation strategy. Is this inspired by another mixed-precision work (in which case, please cite). Have you tried less-heuristic, more systematic alternatives for reallocation?

* Line 184:  “Leading to inaccurate quantization statistics …” Do you have measurements (e.g., per-timestep activation histograms, KL/TV distances) that quantify the mismatch between AR-style calibration and masked-denoising activations?

---

> ### Author Response · Authors · 2025-11-23
> **Response to Reviewer xcW1 (denoted as R3)**
>
> `Q3-1:`Limited analysis of runtime/latency and kernel costs. Quant-dLLM reports a favourable model size (Table 3) but lacks end-to-end latency/throughput on full models and a cost breakdown for DAQ/ABMP kernels. Reporting wall-clock metrics on common hardware and the proportion of time in quant ops would help assess deployability
>
> `A3-1:`Thank you for highlighting the need for runtime and throughput analysis. We provide both the inference latency and the quantization cost to more clearly assess deployability.
>
> For inference, we evaluate the **latency (ms)** of major linear layers in LLaDA-8B-Base using the BitBLAS [ref1] backend. Experiments are conducted on an NVIDIA A6000 with a sequence length of 2048. Our 2-bit model consistently improves throughput over FP16, especially on large feed-forward and output-projection layers:
>
> | Model | Weight Size | FP16 | Ours (2-bit) |
> |----------|--------------|-------|---------------|
> | LLaDA-8B-Base | 4096×4096     | 0.76595 | 0.71948 |
> | LLaDA-8B-Base | 4096×12288    | 1.82547 | 0.83091 |
> | LLaDA-8B-Base | 12288×4096    | 1.97524 | 0.84795 |
> | LLaDA-8B-Base | 4096×126464   | 18.78716 | 8.55148 |
>
> 1. Our 2-bit operator achieves **2.2–2.3× speedups** on the large feed-forward and output-projection layers.
> 2. Even on smaller attention projections like 4096×4096, our method shows slightly lower latency than FP16.
> 3. These results demonstrate **consistent and meaningful speed improvements**, while retaining a deployment-friendly low-bit representation.
>
> For the quantization process, using 128 calibration samples to quantize LLaDA-8B-Base on a single NVIDIA A6000 (48 GB) requires about 37 GB of memory and completes in 32 minutes.
>
> [ref1] https://github.com/microsoft/BitBLAS
>
> `Q3-2:`Sensitivity to hyperparameters is only partially explored: λ (importance mask weight), 3σ thresholding, group/block sizes, or order K choices beyond the {1,2,3} scheme.
>
> `A3-2:`Thank you for the suggestion. We provide additional sensitivity studies on λ, block size, and the 3σ threshold. Across all these dimensions, Quant-dLLM shows consistently robust behavior. Both λ and the 3σ mask yield stable performance within a broad range, and the block-size sweep shows smooth variation without abrupt degradation, indicating that the default choice of 128 is well-balanced. Implementing 2-bit quantization requires retaining enough 1-bit through bit allocation, which makes very high-order K challenging. However, exploring a larger K is a promising direction, but the current setting already provides a strong and balanced trade-off. All results below are reported on LLaDA-8B-Base on ARC-C.
>
> | block size | ARC-C ↑ |
> |------------|---------|
> | fp16       | 0.4411  |
> | 32         | 0.4019  |
> | 64         | 0.3771  |
> | 128 (ours) | 0.3626  |
> | 256        | 0.3311  |
>
> | λ          | ARC-C ↑ |
> |------------|---------|
> | fp16       | 0.4411  |
> | 1          | 0.3413  |
> | 2          | 0.3512  |
> | 3 (ours)   | 0.3626  |
> | 4          | 0.3541  |
> | 5          | 0.3543  |
>
> | σ          | ARC-C ↑ |
> |------------|---------|
> | 4σ         | 0.3603  |
> | 3σ (ours)  | 0.3626  |
> | 2σ         | 0.3614  |
> | 1.5σ       | 0.3532  |
> | 1σ         | 0.3456  |
>
> | Bit-width | Method | ARC-C ↑ |
> |-----------|--------|---------|
> | 16-bit    | FP16   | 0.4411  |
> | 3-bit     | GPTQ   | 0.4322  |
> | 3-bit     | Ours   | 0.4326  |
>
> `Q3-3:`Section 4.4: any discussion or latency/throughput? Are there any costs associated with the quantization method that would affect deployment, e.g., additional kernels, etc?`
>
> `A3-3:`Please check the reply for Q3-1.
>
> `Q3-4:`Ablation studies: I would like to see some ablation studies (even in the appendix) on the effect of the quantization block size or the effect of lamda?
>
> `A3-4:`Thank you for the suggestion. We have added ablations on both block size and λ. For block size, smaller groups provide finer-grained allocation and achieve higher accuracy, but they require storing more scaling parameters. Our choice of 128 follows the standard setting like GPTQ, and offers a practical balance between accuracy and overhead. For λ, it yields stable performance within a broad range and surpasses the baselines significantly.
> The results are shown in A3-2.

---

> ### Author Response · Authors · 2025-11-23
> **Response to Reviewer xcW1 (denoted as R3)**
>
> `Q3-5:`Line 320: regarding the bit relocation strategy. Is this inspired by another mixed-precision work (in which case, please cite). Have you tried less-heuristic, more systematic alternatives for reallocation?
>
> `A3-5:`Our bit relocation strategy is inspired by the mixed-precision design in SliM-LLM, which is among the first to perform group-wise bit allocation on LLMs. Their method uses a double-pointer search to optimize KL-divergence–based allocation, which achieves strong results but introduces additional computation due to iterative search. In contrast, our approach relies on a single salience-based sorting step, which is lighter while still providing competitive performance for dLLMs. Actually, our Hessian-guided formulation brings a degree of systematic structure, but developing a more principled bit-reallocation mechanism remains a promising direction for future work.
>
> `Q3-6:`Line 184: “Leading to inaccurate quantization statistics …” Do you have measurements (e.g., per-timestep activation histograms, KL/TV distances) that quantify the mismatch between AR-style calibration and masked-denoising activations?
>
> `A3-6:`Thank you for the question. To quantify the mismatch between AR-style calibration and masked-denoising activations, we compute cosine similarities across different mask ratios, and compare them with our Masked Calibration Simulation (MCS). Higher similarity indicates closer alignment. In the table below, 0 represents no masking (AR-style) and 1 represents full masking.
>
> | Mask Ratio | 0 | 0.2 | 0.4 | 0.6 | 0.8 | 1.0 |
> |------------|------|------|------|------|------|------|
> | **0 (AR-style)** | 1 | 0.907487 | 0.789597 | 0.649036 | 0.431347 | 0.274203 |
> | 0.2 | 0.907487 | 1 | 0.871791 | 0.719285 | 0.484445 | 0.316993 |
> | 0.4 | 0.789597 | 0.871791 | 1 | 0.828112 | 0.564978 | 0.379756 |
> | 0.6 | 0.649036 | 0.719285 | 0.828112 | 1 | 0.690399 | 0.475251 |
> | 0.8 | 0.431347 | 0.484445 | 0.564978 | 0.690399 | 1 | 0.707796 |
> | 1.0 | 0.274203 | 0.316993 | 0.379756 | 0.475251 | 0.707796 | 1 |
> | **MCS (ours)** | 0.890941 | 0.911415 | 0.901780 | 0.873506 | 0.699994 | 0.651291 |
>
> These results show that AR-style activations drift sharply as the mask ratio increases, making them a poor proxy for masked-denoising behavior. In contrast, MCS maintains much higher similarity at every mask level, demonstrating that it provides a more faithful approximation of the activation statistics used during diffusion decoding.

---

### Official Review · Reviewer_skef · 2025-11-03

**Soundness:** 3
**Presentation:** 1
**Contribution:** 3
**Rating:** 6
**Confidence:** 3

**Summary:**

The paper proposes 2-bit weight-only quantization methods tailored for masked diffusion language models (MDLMs). To address the limitations of GPTQ on MDLMs, it introduces three techniques: MCS, which corrects statistics during masked denoising; DAQ, which matches (WX) instead of (W) by using data/activations (X) during quantization; and ABMP, which assigns 3-bit to top-K sensitive blocks and 1-bit to bottom-K (best around 5–10%). The improvement on standard benchmarks is significant.

**Strengths:**

The method effectively prevents GPTQ from breaking down at 2-bit extreme quantization. Accuracy remains high enough to maintain, to some extent, performance on complex tasks such as math and coding, where 2-bit GPTQ mostly fails. Empirical memory saving exceeds 4× for an 8B MDLM.

**Weaknesses:**

* Comparisons are limited: no low-rank adapters, QAT, hybrid weight+activation quantization.
* The writing needs overall improvement and simplification.
* The naming of sub-methods should be improved: there are too many acronyms, and several are grammatically awkward (missing nouns, missing hyphens, missing capitalization after hyphens, “order-**K**”, Quantiz**ation**, etc.). The lowercase “d” in dLLM is also confusing. It is recommended to consult an LLM.
* The extra “=0” at the end of each function should be removed when compiling the algorithm environment.
* The full-precision model is missing in the math and coding bar plots.
* Although significantly better than GPTQ, the performance drop is still notable, leaving room for improvement.
* Empirical latency/throughput is not reported.
* K is hand-tuned but not heuristic.
* Activation statistics are not reported, so the effect of the correction is invisible.

**Questions:**

* I wonder if SVDQuant [1] (with native low-bit CUDA kernels, activation quantization) could be combined with the proposed method?
* Does the distribution mismatch in MDLMs arise from the increasing number of “MASK” tokens over timesteps? If so, could this be partially resolved by [2]?

[1] Li, et al. “SVDQuant: Absorbing Outliers by Low-Rank Component for 4-Bit Diffusion Models.” ICLR 2025. [arXiv:2411.05007](https://arxiv.org/pdf/2411.05007).

[2] Deschenaux, et al. “Partition Generative Modeling: Masked Modeling Without Masks.” [arXiv:2505.18883](https://arxiv.org/pdf/2505.18883).

---

> ### Author Response · Authors · 2025-11-23
> **Response to Reviewer skef (denoted as R2)**
>
> `Q2-1:`Comparisons are limited: no low-rank adapters, QAT, hybrid weight+activation quantization.
>
> `A2-1:`Thank you for the suggestion. Our method focuses on weight-only PTQ, and is therefore orthogonal to low-rank adaptation techniques. To the best of our knowledge, there are currently no low-rank adapters, QAT, or joint weight–activation quantization methods specifically designed for diffusion-based LLMs. Expanding Quant-dLLM to such hybrid or QAT frameworks is an interesting direction for future work.
>
> `Q2-2:`The writing needs overall improvement and simplification.
>
> `A2-2:`We apologize for the issues in clarity and organization. We will carefully revise the writing in the last version to improve readability and presentation, and make the key ideas easier to follow.
>
> `Q2-3:`The naming of sub-methods should be improved: there are too many acronyms, and several are grammatically awkward (missing nouns, missing hyphens, missing capitalization after hyphens, “order-K”, Quantization, etc.). The lowercase “d” in dLLM is also confusing. It is recommended to consult an LLM.
>
> `A2-3:`Thank you for pointing this out, and we apologize for the inconsistent naming. We will revise the acronyms to ensure clearer grammar, proper capitalization, and more consistent terminology. For the lowercase “d” in dLLM, we followed the usage in prior efficient diffusion-LLM papers such as Quantization Meets dLLMs[ref1] and Fast-dLLM[ref2], but we agree it may be confusing and will clarify this choice in the revised version.
> [ref1] https://arxiv.org/abs/2508.14896
> [ref2] https://arxiv.org/abs/2505.22618
>
> `Q2-4:`The extra “=0” at the end of each function should be removed when compiling the algorithm environment.
>
> `A2-4:`We apologize for the formatting issue. We will remove the stray “=0” in the revised version to ensure the algorithm environment compiles correctly.
>
> `Q2-5:`The full-precision model is missing in the math and coding bar plots.
>
> `A2-5:`We apologize for the omission. We will update the math and coding bar plots in the revised version to include the full-precision model for a complete comparison.
>
> `Q2-6:`Although significantly better than GPTQ, the performance drop is still notable, leaving room for improvement.
>
> `A2-6:`Thank you for the comment. We agree that 2-bit precision has intrinsic expressive limitations, and improving performance at this extreme compression level is an important direction for future work. Our framework, however, also scales to higher bit-widths. At 3-bit, Quant-dLLM already achieves accuracy very close to full precision and slightly better than GPTQ:
>
> | Bit-width | Method | ARC-C ↑ |
> |----------|--------|---------|
> | 16-bit   | FP16   | 0.4411  |
> | 3-bit    | GPTQ   | 0.4322  |
> | 3-bit    | Ours   | 0.4326  |
>
> These results show that while 2-bit is challenging, Quant-dLLM performs strongly at higher bit-widths and approaches full-precision quality already at 3-bit.
>
> `Q2-7:`Empirical latency/throughput is not reported.
>
> `A2-7:`Thank you for raising the question about throughput. We acknowledge that evaluating runtime performance is crucial for demonstrating the practical feasibility of our proposed implementation. Unfortunately, previous works such as Slim-LLM did not report runtime performance due to the lack of a CUDA kernel for matrix multiplication between FP activation and 1-bit weights. We use the BitBLAS [ref3] codebase to benchmark our method, providing detailed runtime evaluations. We evaluate the runtime inference metrics by measuring the latency (ms) of various linear layers in LLaDA-8B-Base. The sequence length of input tensor X is 2048, and experiments are conducted on an NVIDIA A6000 GPU.
>
> | Model | Weight Size | FP16 | Ours (2-bit) |
> |----------|--------------|-------|---------------|
> | LLaDA-8B-Base | 4096$\times$4096 | 0.76595 | 0.71948 |
> | LLaDA-8B-Base | 4096$\times$12288 | 1.82547 | 0.83091 |
> | LLaDA-8B-Base | 12288$\times$4096 | 1.97524 | 0.84795 |
> | LLaDA-8B-Base | 4096$\times$126464 | 18.78716 | 8.55148 |
>
> 1. Our 2-bit operator shows significant acceleration over FP16, reducing the latency of large feed-forward and output-projection layers by 2.2–2.3×. Even for the smaller 4096$\times$4096 attention projections, our method achieves slightly lower latency than FP16.
>
> 2. These results demonstrate that our 2-bit quantization provides consistent and meaningful speed improvements across all major linear layers, while maintaining a deployment-friendly weight representation.
>
> [ref3] https://github.com/microsoft/BitBLAS
>
> `Q2-8:` K is hand-tuned but not heuristic.
>
> `A2-8:`Thank you for the clarification. In our method, K is the representation order rather than a tunable hyperparameter; it directly corresponds to the bit-width assigned by ABMP and is determined by the fixed 2-bit budget, not chosen heuristically.

---

> ### Author Response · Authors · 2025-11-23
> **Response to Reviewer skef (denoted as R2)**
>
> `Q2-9:`Activation statistics are not reported, so the effect of the correction is invisible.
>
> `A2-9:`Thank you for pointing this out. To make the effect of our correction visible, we report **cosine similarities** between activations under different mask ratios, and compare them with the cosine similarity between those AR-style activations and our MCS activations. Here, 0 means no masking (AR-style calibration) and 1 means fully masked.
>
> | Mask Ratio | 0 | 0.2 | 0.4 | 0.6 | 0.8 | 1.0 |
> |------------|------|------|------|------|------|------|
> | **0 (AR-style)** | 1 | 0.907487 | 0.789597 | 0.649036 | 0.431347 | 0.274203 |
> | 0.2 | 0.907487 | 1 | 0.871791 | 0.719285 | 0.484445 | 0.316993 |
> | 0.4 | 0.789597 | 0.871791 | 1 | 0.828112 | 0.564978 | 0.379756 |
> | 0.6 | 0.649036 | 0.719285 | 0.828112 | 1 | 0.690399 | 0.475251 |
> | 0.8 | 0.431347 | 0.484445 | 0.564978 | 0.690399 | 1 | 0.707796 |
> | 1.0 | 0.274203 | 0.316993 | 0.379756 | 0.475251 | 0.707796 | 1 |
> | **MCS (ours)** | 0.890941 | 0.911415 | 0.901780 | 0.873506 | 0.699994 | 0.651291 |
>
> These results show that activations vary substantially across mask ratios, indicating that AR-style calibration cannot capture the distribution shift introduced by masked-denoising. In contrast, our MCS activations maintain consistently high cosine similarity across all mask ratios, providing a more stable match to the full family of masked activations.
>
> `Q2-10:` I wonder if SVDQuant [1] (with native low-bit CUDA kernels, activation quantization) could be combined with the proposed method?
> [1] Li, et al. “SVDQuant: Absorbing Outliers by Low-Rank Component for 4-Bit Diffusion Models.” ICLR 2025.
> https://arxiv.org/pdf/2411.05007
>
> `A2-10:`Thank you for the constructive suggestion. I believe SVDQuant can indeed be combined with Quant-dLLM. In SVDQuant, the residual weights after outlier absorbing are currently quantized with INT4. These INT4 quantizers can be replaced by the proposed DAQ (Data-aware Any-order Quantizer) in Quant-dLLM to further compress the residual branch. Since the residual after outlier absorbing is easier to quantize, DAQ is expected to push it down to 2-bit while maintaining quality, thereby reducing the memory footprint. Integrating SVDQuant’s decomposition with our DAQ quantization pipeline appears feasible; however, supporting SVDQuant’s custom CUDA kernels would require future system-level work.
>
> `Q2-11:`Does the distribution mismatch in MDLMs arise from the increasing number of “MASK” tokens over timesteps? If so, could this be partially resolved by [2]?
> [2] Deschenaux, et al. “Partition Generative Modeling: Masked Modeling Without Masks.
> https://arxiv.org/pdf/2505.18883
>
> `A2-11:`Thank you for the thoughtful question. The distribution mismatch in MDLMs is indeed related to the growing proportion of [MASK] tokens across timesteps, which causes the activation distribution during diffusion decoding to drift away from the AR-style activations typically used for PTQ calibration.
>
> Reference [2] proposes a masking-free training strategy that may reduce this shift at the architectural or pretraining level, but it requires changing the model’s training objective and inference rule. By contrast, Quant-dLLM is a training-free, architecture-preserving PTQ method, and cannot rely on such modifications.
>
> To address the mismatch within the PTQ setting, our Masked Calibration Simulation (MCS) explicitly simulates timestep-aware masked activations. As shown by the cosine-similarity measurements (see A2-9), AR-style activations diverge sharply from masked activations as the mask ratio increases, while MCS maintains much higher similarity across all mask levels. This demonstrates that MCS partially reduces the distribution mismatch without altering the model or adding computational overhead.

---

> > ### Comment · Reviewer_skef · 2025-11-27
> >
> > Thanks for your responses.
> > To clarify on 2, there is a missing noun after "Adaptive Blockwise Mixed Precision"; "Precision" does not refer to a module. "Data-aware Any-order Quantizer" should be "Quantization" (or otherwise "Masked Calibration Simulation" should be "Simulator", either way).
> > To clarify on 11, the question was about whether Quant-dLLM will work better on the PGM architecture.

---

> > > ### Author Response · Authors · 2025-12-02
> > >
> > > Thank you very much for your follow-up, your clarifications, and your time.
> > > For point (2), we agree that “Precision” alone does not denote a module. In the final version, we will make the three components grammatically parallel and module-like.
> > > For point (11), we appreciate the clarification that the question is about the PGM architecture. Conceptually, Quant-dLLM can transfer to PGMs: DAQ and ABMP are architecture-agnostic and only rely on layer-wise activations, while MCS can be adapted by sampling PGM-style partitions and timesteps instead of explicit Bernoulli masks. Since PGMs avoid carrying many [MASK] tokens and have more stable denoising dynamics than MDLMs, we expect Quant-dLLM to remain effective and potentially become easier to apply.

---

### Official Review · Reviewer_dsVf · 2025-11-21

**Soundness:** 4
**Presentation:** 3
**Contribution:** 3
**Rating:** 8
**Confidence:** 3

**Summary:**

The paper correctly highlights the timestep-dependent masking in dLLMs and the accumulated denoising-step error, both of which invalidate classic AR-LLM PTQ assumptions (page 1–2). This motivation is strong and well supported by experimental trends.

**Strengths:**

The DAQ method builds on RC-scaled binarization and extends it to multi-order binary compositions (Algorithm 2, page 5).
Strong points:
Data-aware objective reformulation using second-moment statistics (Eq. 4).
Importance mask using 3\sigma outlier detection (page 5) is computationally cheap.
Closed-form alternating updates (Eqs. 8–9) are well derived.
Multi-order fitting of residuals is natural and preserves binary-friendly execution.
The ablations (Table 2c) show DAQ contributes the largest accuracy jump.

Across 5 dLLMs and 7 tasks, the improvement over GPTQ/GPTAQ/Slim-LLM is consistent and large (Table 1, page 7).

Performance remains robust even on math and code tasks, where low-bit AR PTQ collapses (Figure 3, page 8).

**Weaknesses:**

The 3σ rule (page 5) is heuristic. Can the authors give a intuition or analysis on:
-stability of \DELTA across calibration seeds,
- sensitivity of performance to \sigma threshold or \lambda,
- whether importance correlates with diffusion-step Jacobians or outlier distributions

Group size = 128 is used universally. No analysis on: effect of block size, potential misalignment between block boundaries and meaningful weight structure.

**Questions:**

Can the authors comment on sacalability? i.e. scaling to larger models (≥30B) and comment on the generality claim.

---

> ### Author Response · Authors · 2025-11-23
> **Response to Reviewer dsVf (denoted as R1)**
>
> `Q1-1:`The 3σ rule (page 5) is heuristic. Can the authors give a intuition or analysis on: stability of Δ across calibration seeds, sensitivity of performance to σ threshold or λ, whether importance correlates with diffusion-step Jacobians or outlier distributions.
>
> `A1-1:`Thank you for the question. The importance mask is derived from the MCS-based Z-matrix, whose heavy-tailed structure remains stable across calibration seeds. Our ablations further show that the method is robust to both σ and λ choices: across all reasonable settings, Quant-dLLM consistently achieves strong performance and remains far above baseline PTQ methods. Below we provide the results on LLaDA-8B-Base on ARC-C.
>
> | λ | ARC-C ↑ |
> |----|--------|
> | fp16 | 0.4411 |
> | 1 | 0.3413 |
> | 2 | 0.3512 |
> | 3 (ours) | 0.3626 |
> | 4 | 0.3541 |
> | 5 | 0.3543 |
>
> | σ | ARC-C ↑ |
> |------|--------|
> | fp16 | 0.4411 |
> | 4σ | 0.3603 |
> | 3σ (ours) | 0.3626 |
> | 2σ | 0.3614 |
> | 1.5σ | 0.3532 |
> | 1σ | 0.3456 |
>
> The importance matrix Z highlights activation directions that repeatedly exhibit large variance across diffusion timesteps—patterns that align with high local Jacobian sensitivity and heavy-tailed weight outliers. This yields a stable and meaningful salience structure, contributing to the robustness observed in the above ablations.
>
> `Q1-2:`Group size = 128 is used universally. No analysis on: effect of block size, potential misalignment between block boundaries and meaningful weight structure.
>
> `A1-2:`Thank you for the question. We follow the standard choice of block size = 128, which is widely adopted in prior PTQ methods such as GPTQ [ref1]. To verify this setting, we include ablations below. Smaller blocks provide finer-grained allocation and improve accuracy, but they require storing more scaling parameters. Block boundaries may indeed, to some extent, fail to align with meaningful weight structure. But，introducing dynamic block sizes or redefining blocks would add nontrivial computation, making it an interesting direction for future work. Overall, the performance varies smoothly across block sizes, and 128 offers a practical balance between accuracy and overhead.
>
> | block size | ARC-C ↑ |
> |------------|--------|
> | fp16 | 0.4411 |
> | 32 | 0.4019 |
> | 64 | 0.3771 |
> | 128 (ours) | 0.3626 |
> | 256 | 0.3311 |
>
> [ref1] https://arxiv.org/abs/2210.17323
>
> `Q1-3:`Can the authors comment on sacalability? i.e. scaling to larger models (≥30B) and comment on the generality claim.
>
> `A1-3:`Thank you for the question. Our quantization procedure operates layer by layer, so it scales naturally to larger models in the ≥30B range. The main computations depend only on the size of each individual layer, which makes the method scalable. At present, there are no open-source dLLMs larger than 30B, so we unfortunately could not include experiments at that scale.

---

### Comment · Area_Chair_Hq2m · 2025-11-27
**Please take a moment to read the authors’ responses**

Dear Reviewers,

I hope this message finds you well. As the Area Chair for this submission, I would like to kindly remind you that the author rebuttal is now available. Please take a moment to read the authors’ responses and, if necessary, update your reviews accordingly.

Thank you very much for your time and for contributing to the quality of ICLR 2026.

Best regards,

Your AC

---

### Author Response · Authors · 2025-12-02
**Summary of rebuttal and discussion (Part 1)**

Dear Area Chair and Reviewers,

We sincerely appreciate all Reviewers for their thoughtful evaluations and constructive feedback. We are encouraged by the strong and consistent recognition of our work’s novelty, technical soundness, and empirical advantages. Below, we summarize the major strengths acknowledged across the reviews.

1.**Innovation & Novelty.**
Reviewers found our work to be one of the first that directly targets the unique characteristics of diffusion LLMs. Reviewer xcW1 noted that the paper "identifies the idiosyncrasies of diffusion LLMs (dLLMs)" and "addresses a real gap between autoregressive and diffusion-style inference". Reviewer ayMd described our work as "one of the pioneering studies" that provides meaningful guidance for future research. Reviewer hDfx highlighted that "MCS directly tackles the distributional mismatch caused by masked denoising in dLLMs". These comments show clear recognition of the novelty and impact of our contribution.

2.**Technical Soundness.**
Reviewers appreciated the clarity and correctness of our method. Reviewer dsVf emphasized the strength of the data-aware reformulation, the efficiency of the 3σ importance mask, and the closed-form alternating updates. Reviewer xcW1 found the approach "conceptually elegant and technically well-executed." Reviewer hDfx stated that the paper is "well-written, with detailed algorithms and clear exposition." These remarks show that the method is solid, well-motivated, and clearly presented.

3.**Empirical Performance.**
Reviewers consistently praised our strong and stable results. Reviewer dsVf noted "consistent and large" improvements across 5 dLLMs and 7 tasks, including math and code, where low-bit AR PTQ collapses. Reviewer skef stated that our method prevents GPTQ from breaking down at 2-bit and maintains accuracy. Reviewer xcW1 also highlighted the "consistent improvements across all benchmarks." Reviewer hDfx confirmed gains on challenging math and code tasks and emphasized that the ablations verify the contribution of each module. Together, these reviews confirm that Quant-dLLM performs reliably and outperforms existing PTQ baselines.

4.**Practical Value & Efficiency.**
Reviewers also recognized the practical significance of our design. Reviewer skef noted that our approach achieves more than 4× empirical memory saving while maintaining accuracy, and Reviewer xcW1 emphasized that the method supports aggressive compression under a strict 2-bit budget. Reviewer ayMd acknowledged that our work directly addresses the shortcomings of existing quantization methods on dLLMs.

5.**Clarity.**
Reviewers appreciated the clear exposition and the complete experiments. Reviewer hDfx described the paper as "well-written, with detailed algorithms and clear exposition". Reviewer xcW1 noted that the empirical analysis is careful. These comments show that the paper is easy to read and understand, and that the results are straightforward to reproduce.

**Summary of Concerns and Our Responses**

1.**Request for More Experiments and Results.**
Both 5 reviewers requested more empirical evidence. This included ablations (λ, σ, block size, K), activation-mismatch analysis, runtime/latency, quantization cost, and representative 2-bit PTQ baselines such as QuIP and PB-LLM.
We added all the requested results:
- λ, σ, and block-size sweeps
- QuIP and PB-LLM 2-bit baselines
- latency measurements via BitBLAS (showing 2.2–2.3× speedups)
- PTQ memory/runtime (37 GB, 32 minutes)

2.**Writing, Acronyms, and Terminology.**
Reviewers skef and ayMd found that some parts lack clarity, acronyms were inconsistent, and the term “block” was ambiguous. We will revise the writing, unify acronyms, and clarify that “block” refers to a 128-column quantization group.

3.**Scalability and Generality.**
Reviewer dsVf asked about scaling to ≥30B models and the generality claim. We clarified that Quant-dLLM quantizes layer-wise and therefore scales naturally. The limitation comes from the absence of ≥30B open-source dLLMs.

4.**Distribution Mismatch and Effectiveness of MCS.**
Reviewers skef and xcW1 requested quantitative evidence for activation mismatch and MCS effectiveness. We added cosine-similarity measurements showing strong mismatch under AR-style calibration and much higher consistency under MCS.

---

### Author Response · Authors · 2025-12-02
**Summary of rebuttal and discussion (Part 2)**

**Summary**

Our paper received generally positive initial scores, with reviewers acknowledging the novelty of studying post-training quantization for diffusion LLMs and the strength of the empirical results. Reviewers dsVf, skef, xcW1, ayMd, and hDfx highlighted three main contributions:
 (1) identifying the unique challenges of quantizing diffusion-style LLMs,
 (2) introducing a technically sound data-aware any-order quantization framework, and
 (3) demonstrating consistent improvements across a wide range of tasks, including difficult math and coding benchmarks.

At the same time, reviewers raised several concerns in the initial round, including requests for more ablations and baselines, clearer runtime and quantization-cost reporting, clarification of writing and terminology, and analysis of the stability of the hyperparameters. These were reasonable points and helped sharpen the focus of the work.

During the rebuttal stage, the authors provided comprehensive responses: new λ/σ/block-size sweeps, activation-similarity analysis, QuIP and PB-LLM 2-bit baselines, BitBLAS-based latency results showing 2.2–2.3× speedups, and full PTQ memory/runtime costs. The authors also clarified naming, block definitions, and distinctions from Slim-LLM. These additions addressed the major concerns raised in the initial reviews.

Reviewers acknowledged the improvements. Reviewer hDfx noted that “the response has alleviated my concerns,” and reviewers skef and ayMd expressed appreciation for the clarifications and updates.

The overall consensus is that the contributions are meaningful, the method is technically sound, and the empirical evidence is now complete and convincing. As several reviewers noted, the paper provides a “pioneering” and “well-executed” framework that advances PTQ for diffusion LLMs. In light of the strengthened experimental support and the reviewers’ updated assessments, the paper is viewed as a solid and impactful contribution.

---

### Meta-Review · Area_Chair_fwYc · 2026-01-04

**Summary:**

This paper received 5 reviews. In general, the reviews are positive. Most of the reviewers acknowledge this pioneering work of studying post-training quantization (PTQ) for the latest foundation models – dLLMs, strong performance and its potential impacts, despite that it is limited to 2-bit weight-only quantization. There were some concerns regarding runtime, sensitivity, etc, which had been well addressed in the rebuttal. The authors also provide the performance on the 3-bit setting in the rebuttal, for which the performance gain is minor, compared to GPTQ. Overall, this is a significant quantization work for dLLMs.

**Reviewer Scores:**

The reviews are reasonable. The only reviewer with a negative rating might consider upgrading their rating, as the rebuttal is quite convincing.

---

### Decision · Program_Chairs · 2026-01-26

Accept (Poster)